

# Landslides, floods and sinkholes in a karst environment: the 1-6 September 2014 Gargano event, southern Italy

Maria Elena Martinotti[1], Luca Pisano[2], Ivan Marchesini[1], Mauro Rossi[1], Silvia Peruccacci[1], Maria Teresa Brunetti[1], Massimo Melillo[1], Giuseppe Amoruso[3], Pierluigi Loiacono[3], Carmela Vennari[2,4],
Giovanna Vessia[2,5], Maria Trabace[3], Mario Parise[2], Fausto Guzzetti[1]

[1]Consiglio Nazionale delle Ricerche, Istituto di Ricerca per la Protezione Idrogeologica, via Madonna Alta 126, I-06128 Perugia, Italy
[2]Consiglio Nazionale delle Ricerche, Istituto di Ricerca per la Protezione Idrogeologica, via Amendola 122, I-70126 Bari, Italy
[3]Regione Puglia, Servizio di Protezione Civile, Via delle Magnolie 6/8, I-70126 Modugno (Bari), Italy
[4]University of Naples "Federico II", Naples, Italy
[5]University of Chieti-Pescara "Gabriele D'Annunzio", Chieti, Italy

*Correspondence to*: Maria Elena Martinotti (maria.elena.martinotti@irpi.cnr.it)

**Abstract.** In karst environments, heavy rainfall is known to cause multiple geo-hydrological hazards, including inundations, flash floods, landslides, and sinkholes. We studied a period of intense rainfall from 1 to 6 September 2014 in the Gargano Promontory, a karst landscape in Puglia, southern Italy. In the period, a sequence of torrential rainfall events caused severe damage, and claimed two fatalities, triggering different types of geo-hydrological hazards. The amount and accuracy of the geographical and the temporal information varied for the different hazards. The temporal information was most accurate for
the inundation caused by a major river, less accurate for flash floods caused by minor torrents, and even less accurate for landslides. For sinkholes, only generic information on the period of occurrence of the failures was available. Our analysis revealed that in the Promontory, rainfall-driven geo-hydrological hazards occurred in response to extreme meteorological conditions, and that the karst landscape responded to the torrential rainfall with a threshold behaviour. We exploited the rainfall and the landslide information to design the new Ensemble – Non Exceedance Probability, E-NEP algorithm for the
quantitative evaluation of the possible occurrence of rainfall-induced landslides, and of related geo-hydrological hazards. The ensemble of the metrics produced by the E-NEP algorithm provided better diagnostics than the single metrics often used for landslide forecasting, including rainfall duration, cumulated rainfall and rainfall intensity. We expect that the E-NEP algorithm will be useful for landslide early warning in karst areas and in other similar environments. We acknowledge that further tests are needed to evaluate the algorithm in different meteorological, geological, and physiographical settings.

## 1 Introduction


Torrential rainfall is known to cause inundations, flash floods, and different types of landslides, including debris flows, soil slides and rock falls. Less known is that intense rainfall can cause sinkholes, a subtle hazard in many karst environments



(Parise and Gunn, 2007; Gutierrez et al., 2014; Parise et al., 2015). Here, we describe a series of rainfall events and their ground effects in the period from 1 to 6 September 2014 in the Gargano Promontory, a karst environment and a popular tourist area in Puglia, southern Italy. In a 6-day period, a sequence of four heavy rainfall events, separated by periods with little or no rainfall, caused multiple geo-hydrological hazards in the Promontory, including landslides, flash floods,

widespread inundation, and sinkholes. The death toll amounted to two fatalities, and a number of people were forced to leave their homes or businesses. Urban areas, tourist resorts, roads and rails were inundated and damaged by landslides, flash floods and inundations, causing severe economic consequences. We used the rainfall and the geo-hydrological hazards information to investigate the spatial-temporal relationships between the rainfall trigger and the geo-hydrological hazards, and to design and test an algorithm for improved early landslide (and possibly other geo-hydrological hazards) warning.

The paper is organized as follows. After a brief description of the study area (Section 2), in Section 3 we present the main meteorological and rainfall characteristic of the heavy rainfall period that has resulted in landslides, flash floods, inundations and sinkholes in the Gargano Promontory, and we investigate the spatial-temporal relationships between the intense rainfall and its ground effects. Next, in Section 4 we present a new method to forecast the possible occurrence of rainfall-induced landslides – and associated geo-hydrological hazards – based on the continuous monitoring of local rainfall conditions. This

is followed, in Section 5, by a discussion of the results obtained, including general considerations on the geo-hydrological effects of intense rainfall in karst environments, and their possible predictability using the new rainfall-based forecasting method. We conclude (Section 6) summarizing the lessons learnt.

**2 Study area**

The study area covers approximately 1,600 km$^2$, and encompasses the Gargano Promontory that extends for a few tens of

kilometres into the Adriatic Sea, in the NE part of the Puglia (Apulia) region, southern Italy (Fig. 1). The Lesina and Varano coastal lakes separate the northern side of the promontory from the sea. Elevation in the area ranges from sea level to 1056 m a.s.l. with a mean value of about 400 m, and morphology is controlled by E-W and NW-SE-trending faults (Funiciello et al., 1988; Brankman and Aydin, 2004). Due to the presence of a well-developed karst environment, surface hydrography is limited to a few, short, ephemeral drainages along the slopes that bound the elevated central plateau, and to the Candelaro

River and minor drainages in the alluvial and coastal plains surrounding the Promontory. In the area crop out sedimentary rocks, chiefly carbonate platform limestone, limited marl, and residual "terra rossa" deposits (Bosellini et al., 1999). Soils are thin or absent and, where present, they are chromic cambisols and luvisols. Yearly cumulated rainfall ranges between 400 and 1200 mm, and mean annual air temperature varies from 10 °C to 17 °C. Climate is Mediterranean to Mediterranean sub-oceanic. July and August are dry, and most of the precipitation falls as rainfall from September to November (Polemio

and Lonigro, 2011). The Promontory hosts the Gargano National Park and a number of towns and villages that collectively represent an important touristic area and a significant economic resource.



Although not particularly frequent or abundant compared to other areas in southern Italy, different geo-hydrological hazards exist in the Gargano Promontory, including landslides, floods, flash floods, and sinkholes. In recent historical times, destructive events occurred on 15 July 1972 (Bissanti, 1972) and 29 July 1976, when the city of Manfredonia, to the S of the Promontory (Fig. 1), suffered inundations, and on 10-12 September 1982, when the town of San Marco in Lamis was hit by

torrential rain. Landslides were reported in 1931, 1935, 1950, 1952, 1962, 1963, 1972, 1996, and 1997, and floods in 1996, 1997, 1998, 2002, 2007 and 2011. The main landslide types are rock falls, topples, and small disrupted rock slides that originate primarily from steep rock slopes. Flash floods and coastal floods occur in response to intense rainfall, but are not very frequent in the historical record. The karst environment favours the formation of sinkholes i.e., karst forms also known as "dolines" (Ford and Williams, 2007), with a local density of up to 100 dolines per square kilometre (Castiglioni and

Sauro, 2000; Parise, 2008; Simone and Fiore, 2014). Sinkholes in the Promontory range in size from small to very small features extending a few tens of square meters, to large and deep features including the "Dolina Pozzatina" with a depth of 100 m and a perimeter of about 1850 m, to large polje, including the Sant'Egedio polje, near San Giovanni Rotondo (Fig. 1).

### 3 Events description

### 3.1 Meteorological settings

The meteorological event that brought torrential rainfall in the Gargano area began on 1 September 2014, when a perturbed nucleus originating from northern Europe moved to lower latitudes and impacted the Italian peninsula, starting from the northern and eastern sectors. In the early afternoon of September 1st, the central and southern parts of the Italian peninsula were first affected (Fig. 2A). Between 2 and 3 September, the low pressure vortex moved towards the Ionian Sea, and then to the Balkans and the Hellenic peninsula. The meteorological situation determined an inflow of perturbed masses of air over

most of the Adriatic regions (Fig. 2B,C). The counter-clockwise circulation affected most of central and southern Italy and persisted until September 6th. Residual perturbed meteorological conditions remained over the southern Italian regions, and in particular those facing the Ionian Sea, with isolated minor precipitations until the late morning of September 7th (Fig. 2D).

### 3.2 Rainfall events

The perturbed meteorological conditions over Italy resulted in torrential precipitation in the Gargano Promontory, with

cumulated rainfall exceeding 500 mm in the 6-day period 1–6 September (Fig. 3). To study the intense rainfall period, we used hourly rainfall records available for 39 rain gauges pertaining to the national network of rain gauges operated in the area by the Italian National Department of Civil Protection and the Puglia Regional Government. Inspection of the rainfall records and of the geographical distribution of the precipitation (Fig. 3) revealed that (i) heavy rainfall persisted for the entire observation period, hitting different parts of the promontory at different times, and that (ii) seven periods could be singled

out, including four rainfall ("wet") periods and three no-rainfall ("dry") periods (Fig. 3). The rainfall periods ranged from 8 to 49 hours, and were separated by dry periods of between 11 and 19 hours (Table 1).





The first rainfall period (I) lasted 8 hours, from 12:00 to 20:00 UTC+2 on 1 September. In this "wet" period, around 50 mm of rain were measured by most of the rain gauges, for an average rainfall intensity of about 6.25 mm h$^{-1}$ (Fig. 3, Table 1). After a period of 19 hours without rainfall (II), the second rainfall period started (III) and was particularly severe in the SW part of the Promontory. About 400 mm were measured at the San Giovanni Rotondo and the San Marco in Lamis rain

gauges, corresponding to an average intensity exceeding 8.2 mm h$^{-1}$, with peak values exceeding 40 mm h$^{-1}$. Relatively smaller amounts of rainfall were recorded at the Cagnano Varano (240 mm, most of which between 05:00 UTC+2 and 12:00 UTC+2 on 4 September) and the Monte Sant'Angelo (200 mm) rain gauges. The Vico del Gargano and the Bosco Umbra rain gauges, located in the NE part of the Promontory, recorded approximately 50 mm of rain in the period (Figs. 1, 3). Following a dry period of 12 hours (IV), rainfall started again on 5 September (V), and this time was most abundant in the

NE part of the Promontory. In a 12-hour period, the Bosco Umbra and Vico del Gargano rain gauges measured slightly more and slightly less than 100 mm of rain, respectively, corresponding to a rainfall intensity exceeding 8.0 mm h$^{-1}$. On the opposite, southern side of the Promontory, the San Marco in Lamis rain gauge recorded only 10 mm of rainfall. In the same period (V), the San Giovanni Rotondo and Monte Sant'Angelo rain gauges, in the SE part of the Promontory, measured about 50 mm of rainfall. Period V was followed by an 11-hour dry period (VI) which ended at 03:00 UTC+2 of 6 September

2014, when intense rainfall started again. The last rainfall period (VII) lasted until 14:00 UTC+2. In the 11-hour period all the considered rain gauges measured more than 50 mm of rain, with the maximum cumulated value recorded by the Vico del Gargano rain gauge, where 140 mm of rain were measured, corresponding to a rainfall intensity exceeding 12.0 mm h$^{-1}$. A rank analysis of rainfall measurements for six rain gauges in the 7-year period from April 2009 to April 2016, highlighted the severity of the 6-day rainfall period (Fig. 4). Except for the Monte Sant'Angelo rain gauge, located in the southern part of

the Promontory, the 1–6 September rainfall period exhibited the highest cumulated rainfall in the observation period. Adopting the classification proposed by Alpert et al. (2002), the rainfall was "torrential" in all the considered rain gauges, and for three of the rain gauges the only "torrential" event in the (short) record available to us (Fig. 4).

### 3.3 Landslides, floods and sinkholes

The sequence of intense rainfall events resulted in a number of geo-hydrological hazards, including floods, flash floods,

landslides and sinkholes, and caused one flood fatality at Peschici, one landslide fatality at Carpino (http://polaris.irpi.cnr.it/), and severe socio-economic damage. Throughout the Promontory, the road and railway networks were interrupted at several sites by inundations (Fig. 6A) and landslides, and many road underpasses were clogged by debris and sediments (Fig. 6B). In several towns and dwellings the inhabitants were evacuated from their homes, and a number of touristic resorts were inundated by water, mud, and debris.

Flooding was widespread in the Candelaro catchment that bounds to the SW the Gargano Promontory (Fig. 5). Two hydrological gauging stations located where the Candelaro River crosses State Road SS 272, W of the Gargano range, and where it crosses the Provincial Road SP 60 near to the outlet in the Manfredonia Gulf (Fig. 5), measured very high water levels. The upstream gauge along the SS 272 measured a first peak of 5.30 m at 02:00 UTC+2 on September 4, followed by





a slightly higher peak of 5.50 m at 06:30 UTC+2. The water level remained very sustained until 09:00 UTC+2, then it diminished (Fig. 3H). At the downstream gauging station located along SP 60, about 30 km downstream form the upstream gauge, a peak of 3.77 m was measured at 16:00 UTC+2 on September 4, about 10 hours later than the peak measured by the upstream gauge. We justify the (significant) time difference in the peak discharge by the widespread inundation of the Candelaro River plain (Fig. 6A).

Inundations were also severe along the northern coastal area, between the Varano Lake and Vieste, and particularly between the towns of Cagnano Varano and Carpino, where flooding caused one fatality (Fig. 5). Near the Varano Lake, large agricultural areas were inundated. Along the northern coast of the Promontory, flash floods produced by small torrents occurred mostly on 5–6 September. In the morning of 5 September, the Macchio Torrent overflowed and inundated Vieste, and several touristic sites. Overflowing of minor torrents and ditches was reported in the early hours of 6 September in the towns of Peschici, Vico del Gargano, and Rodi Garganico.

The torrential rain caused a number of landslides, mostly shallow (Fig. 6C, E, F). Overall, we collected information on 46 landslides, including 15 soil slips, 14 debris flows, 11 soil slides, four rock falls, and two slope failures of undetermined type. This is a subset of all the event landslides in the Gargano Promontory. Based on the type of failures, we hypothesize that all the landslides were from rapid to extremely rapid. Most of the mapped landslides were in the municipalities of San Marco in Lamis, Ischitella, and Cagnano Varano. Landslides were also reported near San Giovanni Rotondo, Rignano Garganico, and Rodi Garganico (Fig. 5).

We searched information on the time, or period of occurrence of the landslides. However, for most of the landslides the time or period of occurrence remains unknown, or suffers from very large uncertainty. For only nine landslides we obtained reasonably accurate information on the period of occurrence of the slope failures. On 3–4 September, four landslides occurred near San Marco in Lamis, along SS 272, most probably in the 3-hour period between 23:00 and 02:00 UTC+2. We consider these landslides representative of a larger cluster of landslides in the same area (cluster A). On 4 September, a landslide occurred between 05:00 and 16:00 UTC+2 near Cagnano Varano (cluster B). In the same day, a single landslide occurred in the municipality of San Giovanni Rotondo at an undetermined time between 14:00 and 21:00 UTC+2 (cluster C). Lastly, three landslides occurred in the night between 5–6 September, and most probably between 23:00 and 05:00 UTC+2, in the municipalities of Ischitella and Rodi Garganico (cluster D). We attribute the scarce temporal information and the poor accuracy of the information on the time of occurrence of the failures to the difficulty to reach some of the places where the landslides occurred, and to the fact that many landslides occurred in the evening or during the night, and were reported only several hours after the event.

The torrential rainfall also caused sinkholes. We mapped ten, small sinkholes near the villages of San Marco in Lamis and Monte Sant'Angelo (Fig. 5). Based on their morphology and shape, the sinkholes were classified as collapse or cover-collapse sinkholes (Gutiérrez et al., 2008, 2014). At San Marco in Lamis, four sinkholes affected the lower part of a pre-existing karst depression. The deepest sinkhole was about 6 m deep, 5 m wide, and exposed limestone and residual "terra rossa" deposits that represent the upper part of the epikarst (Williams, 2008) (Fig. 6D). Other sinkholes were less developed,



and were detected and mapped locally only based upon morphological considerations. Due to the remote areas where the sinkholes occurred, their limited sizes (Fig. 6G), and the difficulty to detect them, no accurate information is available on the time or period of occurrence of the sinkholes.

### 3.4 Spatial and temporal distributions of rainfall and geo-hydrological hazards

To help investigating the effect of the changing spatial and temporal distribution of the rainfall on the location of the event geo-hydrological hazards (landslides, floods, sinkholes), we prepared Fig. 7 that portrays, for each period in the sequence of rainfall events (Section 3.2), maps showing the spatial distributions of the mean rainfall intensity (in mm h$^{-1}$), the cumulated rainfall for the single period and from the beginning of the event (in mm), and the location of the landslides occurred in each period, and in previous periods.

Inspection of Fig. 7 reveals that the total cumulated rainfall, exceeding 100 mm in large parts of the Promontory, was the result of separate rainfall events that hit different parts of the Promontory at different times. The first rainy period (I) was more widespread but characterized by an overall moderate cumulated rainfall not exceeding 50 mm, and rainfall intensity not exceeding 6.2 mm h$^{-1}$. No geo-hydrological hazard (landslide or flood) was reported during the first rainy period. The second (III) and the third (V) rainy periods were more localised, the second in the central part of the Promontory (San Marco in

Lamis and San Giovanni Rotondo), and the third in the NE sector of the Promontory (Ischitella, Vico del Gargano and Vieste), and were both characterized by high values of the cumulated rainfall (exceeding 400 mm in the second and 130 mm in the third period), and of the rainfall intensity (that exceed 8.5 mm h$^{-1}$ for the second period and 10.5 mm h$^{-1}$ for the third period respectively). Landslides were reported in the central (cluster A) and in the northern (cluster B) parts of the Promontory during the second rainfall period (III). In the same period the upstream hydrological gauging station along the

Candelaro River measured high water flows exceeding 5.0 m (Fig. 2), about two hours after the maximum hourly rainfall measured at the San Marco in Lamis rain gauge, at 24:00 UTC+2 on September 3. The landslides occurred in areas where the total event cumulated rainfall exceeded 400 mm (for cluster A) and 200 mm (for cluster B).

In the central part of the Promontory, landslides were also reported (cluster C) during the second dry period (IV). Given the poor temporal accuracy of these landslides (i.e., between 14:00 and 21:00 UTC+2 on 4 September), we cannot exclude that

the failures occurred during the last few hours of the previous rainfall period (III). The hypothesis is supported by the fact that the San Giovanni Rotondo rain-gauge measured rainfall until 15:00 UTC+2 on 4 September (Fig. 2). The last rainfall period (VII) was again characterized by widespread rainfall throughout most of the Promontory with a distinct peak in the NE sector exceeding 130 mm. Where rainfall intensity was particularly intense (> 11 mm h$^{-1}$), in the NE part of the Promontory, between Rodi Garganico and Peschici, landslides were also reported (cluster D).


### 4 Geo-hydrological hazards forecasting algorithm

We used the rainfall records and the geo-hydrological hazard information available for the Gargano event to design and test an algorithm for the possible operational forecasting of rainfall-induced landslides and other geo-hydrological hazards, including flash floods and sinkholes

### 4.1 The E-NEP algorithm

The Ensemble – Non Exceedance Probability, E-NEP algorithm exploits a standard rainfall records obtained by a rain gauge to trace in time the probability of possible landslide occurrence, and of related geo-hydrological hazards. For the purpose, for each time $t_i$, E-NEP calculates the event rainfall duration $D$, and the corresponding event cumulated rainfall $E$, for increasing antecedent periods before $t_i$, from $t_i$ to $t_i + T$, in $d$ time steps, with $T$ the maximum length of the considered antecedent
rainfall period and $d$ is the time step used to increment the duration of the antecedent rainfall period. For each rainfall-duration–event-cumulated-rainfall pair the corresponding non-exceedance probability (NEP) $p(D,E)$ is obtained using the probabilistic approach proposed by Brunetti et al. (2010), and modified by Peruccacci et al. (2012), for the definition of empirical rainfall thresholds for possible landslide occurrence (Guzzetti et al., 2007). The set of the NEP values (i.e., $\{NEP\}=\{p(D,E)\}$) is then used to determine an ensemble of metrics, including the maximum value of the NEP ($NEP_{max}$), the
10th, 25th, 50th, 75th, and 90th percentiles ($NEP_{10}$, $NEP_{25}$, $NEP_{50}$, $NEP_{75}$, $NEP_{90}$), and the rainfall duration ($D_{NEPmax}$) associated to $NEP_{max}$, that collectively are exploited for landslide (and other geo-hydrological hazards) forecasting. The process is repeated at regular time intervals ($z$, where $t_{i+1} = t_i + z$), allowing to follow the temporal evolution of the probability of possible occurrence of rainfall-induced landslides, and of related geo-hydrological hazards.

Figure 8 portrays the logical schema for the E-NEP algorithm, which consists of two nested loops. First, the maximum
length of the considered rainfall period, $T$ (in hours), the time step to increment the duration of the considered antecedent rainfall, $d$ (in hours), and the time interval before the next set of $(D,E)$ pairs is computed, $z$ (in hours), are set to user-defined values. We stress that the three time (duration) variables are independent, with the only constrain that $d \leq T$. Next, the external loop (cyan in Fig. 8) starts, the rainfall duration $D$ is set to $d$, and $\{NEP\}$ is set to null (an empty set). Next, the internal loop (orange in Fig. 8) starts, and for the given rainfall duration $D$, the corresponding cumulated rainfall $E$ is
determined, and the probability of landslide occurrence $p(D,E)$ is computed adopting the method proposed by Brunetti et al. (2010) and Peruccacci et al (2012) and stored in $\{NEP\}$. The rainfall duration $D$ is then incremented ($D = D + d$) and tested to verify if it is larger than $T$. If $D \leq T$, the internal loop is repeated using the current value of $D$ and the corresponding $E$; else ($D > T$) the loop ends, and the ensemble of metrics of $\{NEP\}$ ($NEP_{10}$, $NEP_{25}$, $NEP_{50}$, $NEP_{75}$, $NEP_{90}$, $NEP_{max}$) and the value of $D_{NEPmax}$ are calculated. The external loop is then repeated after the user-defined time interval of $z$ hours.
Figure 9 exemplifies the application of the E-NEP algorithm to a specific rainfall record, at a given time $t_i$ (i.e. the application of the orange internal loop of Fig. 8), for an antecedent period $T = 96$ hours, with $d = 1$ hour. E-NEP calculates the cumulated event rainfall $E$ for increasing durations $D$, from 1 ($t_i$-1) to 96 h ($t_i$-96), every hour (time step $d = 1$). The



individual ($D,E$) pairs are plotted as grey dots in Fig. 9D. For each pair, the corresponding NEP, the non-exceedance probability of possible landslide occurrence, was calculated using the approach proposed by Brunetti et al. (2010) and modified by Peruccacci et al. (2012), and are shown by the blue squares in Fig. 9D. To further clarify the operations performed by the E-NEP algorithm, the bar charts in Fig. 9A,B,C show, for the same time $t_i$, three antecedent rainfall periods

corresponding to durations of $D = 6$ hours (red bars), $D = 16$ hours (green bars), and $D = 44$ hours (yellow bars). The corresponding cumulated event rainfall (red, green, yellow circles) and the associated non-exceedance probability values (red, green, yellow squares), are shown in Fig. 9D. Lastly, the box plot to the right of Fig. 9D portrays (i) the ensemble of metrics of {NEP}: $NEP_{10}$, $NEP_{25}$, $NEP_{50}$, $NEP_{75}$, and $NEP_{90}$, and the maximum value of the non-exceedance probability, $NEP_{max}$. In Fig. 9D, the green square identifies $NEP_{max}$ i.e., the non-exceedance probability corresponding to the most critical

rainfall condition for possible landslide occurrence in the considered rainfall period. The corresponding duration ($D = 16$ hours) represents the $D_{NEPmax}$ value.

### 4.2 Application and discussion of the E-NEP algorithm

We applied the E-NEP algorithm to the 13-day period between 31 August and 12 September that encompasses the entire series of rainfall events that hit the Gargano Promontory (Fig. 2). We applied the algorithm to synthetic hourly rainfall

records reconstructed for the locations of the four spatial-temporal landslide clusters identified in the study area (Fig. 7). To reconstruct the synthetic rainfall records, we interpolated the hourly rainfall measurements obtained by 39 rain gauges in the Gargano Promontory and the surrounding regions (Fig. 3) at the landslide locations. For the purpose, we used a standard inverse weighted distance (IDW) spatial interpolator (Shepard, 1968) to obtain hourly rainfall values on a regular 500 m × 500 m grid. Next, the hourly rainfall grids were sampled at the grid cells selected to represent the four landslide clusters A,

B, C, and D, and the synthetic hourly rainfall time series were reconstructed for each landslide cluster.

For our analysis, and for each landslide cluster, $t_i$ ranged from 31 August, 00:00 UTC+2 to 11 September, 24:00 UTC+2, in regular 1-hour time intervals ($z$), corresponding to a total of 288 time intervals. For each $t_i$, E-NEP computed the $NEP_{10}$, $NEP_{25}$, $NEP_{50}$, $NEP_{75}$, and $NEP_{90}$ percentiles, the $NEP_{max}$, and the corresponding $D_{NEPmax}$.

Results of the analysis are shown in Fig. 10, for the four landslide clusters, where the single plots show, from top to bottom,

the temporal evolution of (i) the measured and the cumulated rainfall, (ii) the $NEP_{10}$, $NEP_{25}$, $NEP_{50}$, $NEP_{75}$, $NEP_{90}$ percentiles (shown in ranges, by two different shades of blue and by the thick blue line) and $NEP_{max}$ (purple line), and (iii) the corresponding $D_{NEPmax}$ (green line). Figure 10 also shows (i) with grey areas, the possible period of occurrence of the landslides, a measure of the uncertainty in the failure occurrence time, and (ii) with a vertical blue line the time of the high peak measured by the Candelaro gauge along SS 272 (Fig. 10A, Fig. 3).

For cluster A, encompassing landslides occurred along State Road SS 272 and SP 48 near San Marco in Lamis, at 12:00 UTC+2 on 1 September a short rainfall burst hit the landslide area and stopped shortly afterward (Fig. 10 A1, I in Fig. 3). In this period, the $NEP_{max}$ increased rapidly to 0.15 and decreased immediately afterwards due to lack of rainfall (Fig. 10 A2). Next, following a dry period of 19 hours (II in Fig. 3), the main rainfall event started at 16:00 UTC+2 on 2 September (Fig.




10 A1, III in Fig. 3). As a result of this second, intense rainfall event all the NEP percentiles increased abruptly and significantly (Fig. 10 A2), with $NEP_{max}$ exceeding 0.99 at 23:00 UTC+2 on 3 September. The landslides of cluster A followed shortly afterward.

Following the landslide occurrence, all the NEP values remained high for 12 hours. When the rainfall stopped, at 14:00
UTC+2 on September 4, the NEP percentiles decreased, beginning with NEP10 and continuing with the other (larger) percentiles. $NEP_{max}$ decreased below 0.25 at 21:00 UTC+2 on September 8. Of interest is the analysis of the temporal trend of $D_{NEPmax}$ (Fig. 10 A3), the rainfall duration corresponding to largest NEP. $D_{NEPmax}$ (i) increased in response to the first rainfall period (I in Fig. 3), (ii) kept rising during the first dry period (II in Fig. 3), (iii) dropped to zero during the second (main) rainfall period, and precisely when the rainfall intensity reached a maximum value of 7 mm h$^{-1}$ (III in Fig. 3), (iv)
increased steadily for 94 hours, and (iv) remained high until almost the end of the considered period.

We observe that landslides in cluster A occurred when the $NEP_{max}$ was close to its maximum possible value ($NEP_{max} = 1$), a very critical condition for possible landslide initiation (Brunetti et al., 2010; Peruccacci et al., 2012). Just before the landslide occurrence (i) $NEP_{50}$ was close to $NEP_{max}$ i.e., the median value was close to the maximum value of the non-exceedance probability, (ii) the inter-percentile ranges $NEP_{10}$-$NEP_{90}$ and $NEP_{25}$-$NEP_{75}$ were narrow, and (iii) there was a sudden increase
of all NEP values, and particularly of the lower percentiles (i.e., $NEP_{10}$, $NEP_{25}$) (Fig. 10 A2). We further observe that landslides in this cluster occurred after $D_{NEPmax}$ had begun to rise following a sudden drop (Fig. 10 A3).

Inspection of the other plots in Figure 10 reveals significant similarities in the temporal evolution of the metrics computed by the E-NEP algorithm for the other three landslide clusters, when compared to the same metrics computed for cluster A. Specifically, (i) all landslides occurred when the $NEP_{max}$ was close to its maximum value, and immediately before landslide
occurrence (ii) $NEP_{50}$ was close to $NEP_{max}$, (iii) the range $NEP_{25}$-$NEP_{75}$ was narrow, (iv) there was a sudden increase of all NEP percentiles (Fig. 10 A2), except $NEP_{10}$ (Fig. 10 C2), and (v) the landslides occurred after the $D_{NEPmax}$ had started to raise following a previous sudden drop. We consider these observations diagnostic of the rainfall conditions that have resulted in landslides (and other geo-hydrological hazards) in the Gargano Promontory in the period 1-6 September 2014

## 5 Discussion

The analysis of the rainfall records and the geo-hydrological hazard information available for the Gargano Promontory rainfall events between 1 and 6 September 2014 (Section 3), and their application to test the Ensemble – Non Exceedance Probability, E-NEP algorithm (Section 4), allows for general and specific considerations.

We first observe that landslides in the four examined clusters occurred for different levels of the cumulated event rainfall, $E$ (Fig. 7, Table 1). We also observe that rainfall intensity was very high in the period of the failures, or immediately before it,
but also that periods of high rainfall intensity were not associated to (known) landslides (see e.g., clusters C and D, Fig. 10 B, C). We conclude that, in the case of the investigated rainfall events, single metrics like the event cumulated rainfall $E$, and the rainfall intensity $I$, were not singularly diagnostic of the rainfall conditions that have resulted in the known landslides.



As discussed in Section 4.2, a number of potentially diagnostic observations drawn from the ensemble of metrics produced by the E-NEP algorithm were common to all the examined landslide clusters, including the facts that (i) all the landslides occurred when $NEP_{max}$ was close to its maximum possible value, and that (ii) shortly before landslide occurrence there was a sudden increase of all NEP percentiles (except $NEP_{10}$ locally, Fig. 10 C2), $NEP_{50}$ was close to $NEP_{max}$, and the interquartile

range $NEP_{25}$-$NEP_{75}$ was narrow. Following landslides occurrence, $NEP_{max}$ remained typically sustained for long periods, but the NEP percentiles dropped more or less rapidly, even where additional rainfall fell in the area. We observe that no information on landslide occurrence was reported when $NEP_{50}$ was low, or very low. A further observation is that landslides occurred shortly after the $D_{NEPmax}$ had started to rise, following a previous drop. A sudden drop of $D_{NEPmax}$ is always related to an increase in $NEP_{max}$ that is determined by an increase in the rainfall intensity. However, a small increase in rainfall

intensity may not be sufficient to cause $D_{NEPmax}$ to drop. We argue that visual analysis of the temporal evolution of $D_{NEPmax}$ can be exploited to provide indications of the rapid change of the possible critical rainfall conditions that may lead to slope failures shortly afterwards.

We conclude that the ensemble of the metrics produced by the E-NEP algorithm provides better diagnostic results than the single metrics often used for landslide forecasting, including rainfall duration, cumulated event rainfall, and rainfall intensity

(Guzzetti et al. 2007). This is visually portrayed in Fig. 10, where the temporal trend of the cumulated rainfall is less diagnostic than the corresponding trends of the $NEP_{max}$ or the $D_{NEPmax}$ in forecasting the periods of landslide occurrence.

We maintain that the E-NEP algorithm is potentially useful for near-real time landslide warning, but we acknowledge that more investigations are required to test the algorithm in different meteorological, geological, and physiographical settings. The sequences of closely spaced rainfall events in the Gargano Promontory covered a long period (6 days), and this made it

particularly well suited for the design and testing of the E-NEP algorithm. The sequence of rainfall events was also the result of a relatively simple meteorological setting. More tests are needed to evaluate the performance of the E-NEP algorithm for shorter and longer rainfall periods, and in different and more complex meteorological conditions.

We stress that the E-NEP algorithm was designed to attempt to forecast rainfall conditions for the possible occurrence of landslides that react rapidly to a rainfall input. These are typically shallow landslides, including soil slips, debris flows, and

rock falls, that involve the soil or the upper and more weathered parts of the bedrock. E-NEP was not designed to attempt to evaluate other landslides that react slowly, or very slowly to a rainfall input, including e.g., deep-seated landslides, shallow landslides in stiff clay. Also, E-NEP was not designed to attempt to predict landslides caused by meteorological triggers other than intense rainfall, including e.g., rain-on-snow events or rapid snow-melt events (Cardinali et al., 2000). However, we expect that E-NEP can be adapted to forecast shallow landslides caused by intense rainfall even in specific, local

conditions (e.g., in areas burnt by wildfires, Cannon et al., 2003, 2010; De Graff et al., 2013; Moody et al., 2013), provided that sufficient information is available to apply the method proposed by Brunetti et al. (2010) and Peruccacci et al. (2012), that lays at the base of E-NEP.

Analysis of the rainfall conditions that have resulted in landslides, flash floods, inundation and sinkholes in the investigated period in the Gargano Promontory revealed that the geo-hydrological hazards occurred in response to extreme rainfall



conditions. This is confirmed by the facts that (i) rainfall was torrential (Alpert et al., 2002) (Fig. 4), and (ii) the geo-hazards – and particularly the landslides – occurred when all the NEP percentiles were close to the possible maximum value of the non-exceedance probability of possible landslide occurrence (Brunetti et al., 2010; Peruccacci et al., 2012) (Fig. 10), which represent very severe rainfall conditions. The available record of historical landslides and floods indicates that these geo-

hydrological hazards are not very frequent or abundant, compared to other areas in southern Italy. We conclude that in the Gargano Promontory meteorologically-driven geo-hydrological hazards occur in response to extreme (i.e., rare) meteorological conditions. For rainfall-driven geo-hazards, the landscape in the Gargano Promontory exhibits a threshold behaviour that can be modelled conceptually by a Heaviside step function (Abramowitz and Stegun, 1972). For light to heavy rainfall events (Alpert et al., 2002) geo-hydrological hazards do not occur or are rare and minor, whereas for heavy-

torrential to torrential rainfall events geo-hydrological hazards are abundant and particularly disruptive. We attribute the threshold-based behaviour to the karst environment that dominates the landscape in the Promontory.

In the karst environment of the Promontory, rainfall infiltration is efficient even for high intensity rainfall rates, and soils are inexistent or thin, except locally in the sinkholes. This limits the occurrence of landslides, except for very intense (i.e., "extreme") rainfall events. High infiltration to shallow depth in the rock mass facilitates the formation of flash floods,

particularly in small catchments, as frequently occurring also in other parts of Apulia (Parise, 2003; Mossa, 2007). In the sinkholes, the presence of residual soils varies largely, depending on the location, size and depth of the sinkholes. Where the infiltration is reduced, partial or total inundation of the sinkholes occurs. These local inundations are difficult to detect because they last only for short periods, and because they often go unnoticed in the rural, scarcely populated landscape of the Promontory.

The torrential rainfall has triggered sinkholes in the Gargano Promontory (Fig. 6D,G). Accurate information on the time or period of occurrence of the sinkholes is not available, and even the simple detection of the sinkholes was hampered by their small size and the remote location of the events. However, sinkholes represent a subtle and serious hazard in the Promontory, and in other karst areas (Parise and Gunn, 2007; Gutierrez et al., 2014; Parise, 2015; Parise et al., 2015), and establishing methods and procedures for their possible forecasting is of primary interest in karst environments. Based on the analysis of

the 1-6 September 2014 Gargano rainfall period, we confirm that in the Promontory, and in similar karst areas, torrential rainfall can trigger sinkholes, and we hypothesise that approaches based on the near-real-time monitoring of rainfall (e.g., the E-NEP algorithm) can be used to forecast the possible occurrence of rainfall-induced sinkholes. We acknowledge that an analysis of a larger number of rainfall-induced sinkhole events is required to test this hypothesis.

## 6 Conclusions

We studied a period of torrential rain between 1 and 6 September 2014 in the Gargano Promontory, Puglia, southern Italy, that caused a variety of geo-hydrological hazards, including landslides, flash floods, inundations, and sinkholes. We obtained information on the location of the events through field surveys and the analysis of anecdotal information obtained from




various sources. The temporal information varied among the hazards. For inundations the time or period of occurrence were known from gauge data (Fig. 3H), and for flash floods from anecdotal sources (Fig. 6A, B). For landslides, the period of occurrence was inferred from anecdotal sources for only nine (out of 46) slope failures, and with significant uncertainty. No information on the time or period of occurrence was available for the sinkholes. We conclude that the ability to obtain

accurate temporal information for the different hazards, an important information to establish and validate early warning systems, depended on the extent and the location of the different hazards. The temporal information was most accurate for flooding along the Candelaro River (Fig. 3H), followed by flash floods and landslides (Fig. 3), and was not available for the sinkholes.

We used the rainfall and the landslide information available to us to design and test the new Ensemble – Non Exceedance

Probability, E-NEP algorithm for the quantitative evaluation of the probability of possible occurrence of rainfall-induced landslides, and of related geo-hydrological hazards (e.g., flash floods, sinkholes).For the investigated rainfall events, the ensemble of the metrics produced by the E-NEP algorithm provided better diagnostics than the single metrics often used for landslide forecasting, including rainfall duration, cumulated rainfall and rainfall intensity (Guzzetti et al., 2007; Brunetti et al., 2010; Peruccacci et al., 2012). We maintain that the E-NEP algorithm is potentially useful for landslide early warning,

but we acknowledge that more work is needed to test the algorithm in different meteorological, geological, and physiographical settings.

Our analysis revealed that in the Gargano Promontory meteorologically-driven geo-hydrological hazards occur in response to extreme (i.e., rare) meteorological conditions, and the karst landscape responds to torrential rainfall with a threshold behaviour. For light to heavy rainfall events (Alpert et al., 2002) geo-hydrological hazards do not occur or are rare and

minor, whereas for heavy-torrential to torrential rainfall events geo-hydrological hazards are abundant and particularly disruptive, as it was the case for the 1-6 September 2014 event. We maintain that this information is useful for landslide (and other geo-hydrological hazards) early warning systems.

**7 Acknowledgements**

Work performed in the framework of projects supported by the Italian National Department for Civil Protection (DPC), and

the Puglia (Apulia) Regional Government (PRG). Maria Elena Martinotti and Massimo Melillo were supported by two grants of DPC. Luca Pisano was supported by a grant of PRG.

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





**Table 1: Characteristics of seven periods in the sequence of rainfall events that hit the Gargano Promontory between 1 and 6 September 2014. Start and End time is given in UTC+2. Rain/Dry lists rainfall, "wet" (R), and no-rainfall, "dry" (D), periods. Rainfall gives the range (minimum-maximum) of the cumulated rainfall in each period. Intensity is the average intensity in the period for the maximum cumulated rainfall. See also Fig. 3 and Fig. 7.**

| Period | Start day, hour | End day, hour | Length Hour | Rain/Dry | Rainfall mm | Intensity mm h$^{-1}$ |
|---|---|---|---|---|---|---|
| I | 1 Sep, 12:00 | 1 Sep, 20:00 | 8 | R | 20-50 | 6.25 |
| II | 1 Sep, 20:00 | 2 Sep, 15:00 | 19 | D | 0-6 | 0.31 |
| III | 2 Sep, 15:00 | 4 Sep, 16:00 | 49 | R | 50-440 | 8.97 |
| IV | 4 Sep, 16:00 | 5 Sep, 04:00 | 12 | D | 0-6 | 0.50 |
| V | 5 Sep, 04:00 | 5 Sep, 16:00 | 12 | R | 10-130 | 10.83 |
| VI | 5 Sep, 16:00 | 6 Sep, 03:00 | 11 | D | 0-20 | 1.82 |
| VII | 6 Sep, 03:00 | 6 Sep, 14:00 | 11 | R | 50-140 | 12.73 |




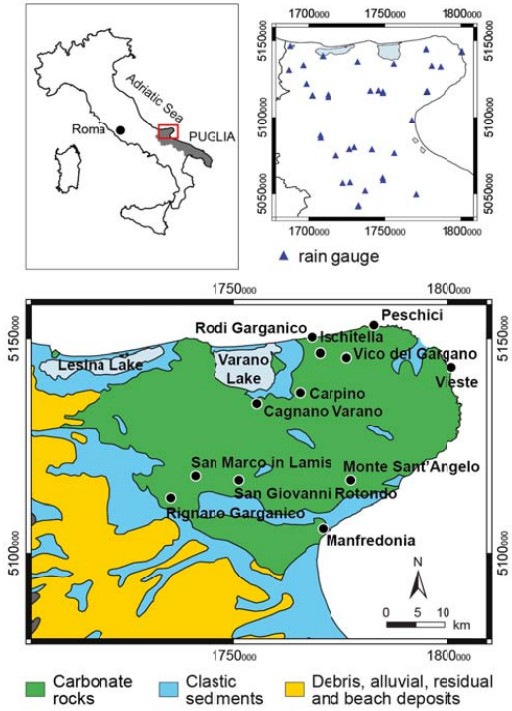

**Figure 1: Upper left map shows location of the study area (red rectangle) in Italy. Grey area is the Puglia (Apulia) region. Upper right map shows location of 39 rain gauges in the study area and the neighbouring area. Lower map shows main rock types (colours) and place names in the study area. WGS84/Pseudo Mercator (EPSG: 3857).**

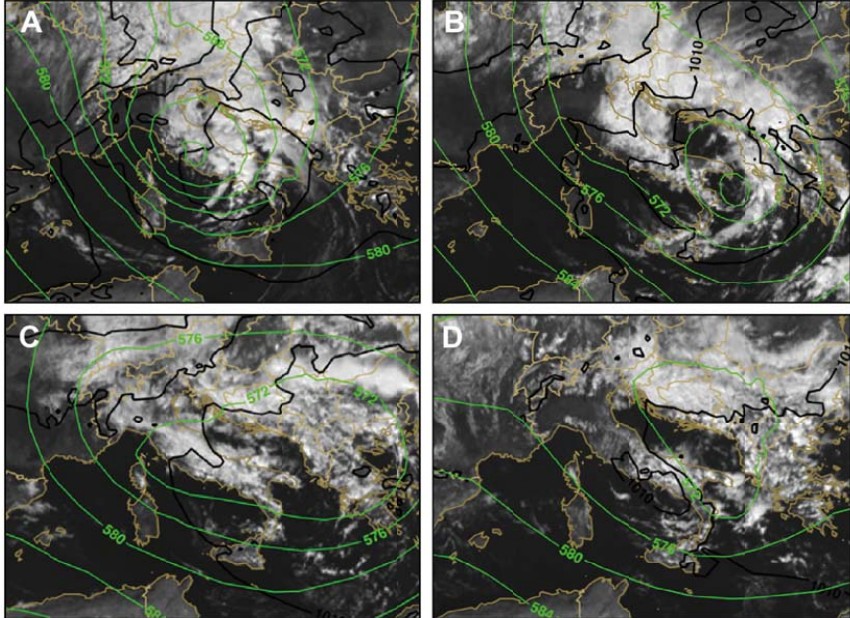

**Figure 2: Meteorological setting for the period 1-6 September 2014 in the Gargano Promontory. Images show Meteosat Second Generation (MSG) – Visible (VIS) 0.6 μm for: (A) 1 September 2014, 12:00 UTC, (B) 2 September 2014, 12:00 UTC, (C) 3 September 2014, 12:00 UTC, (D) 6 September 2014, 12:00 UTC. Green lines show geopotential height of 500 hPa pressure level.**
5   **Black lines show mean seal level pressure. Light brown lines show geographical boundaries. Source: http://www.eumetrain.org.**





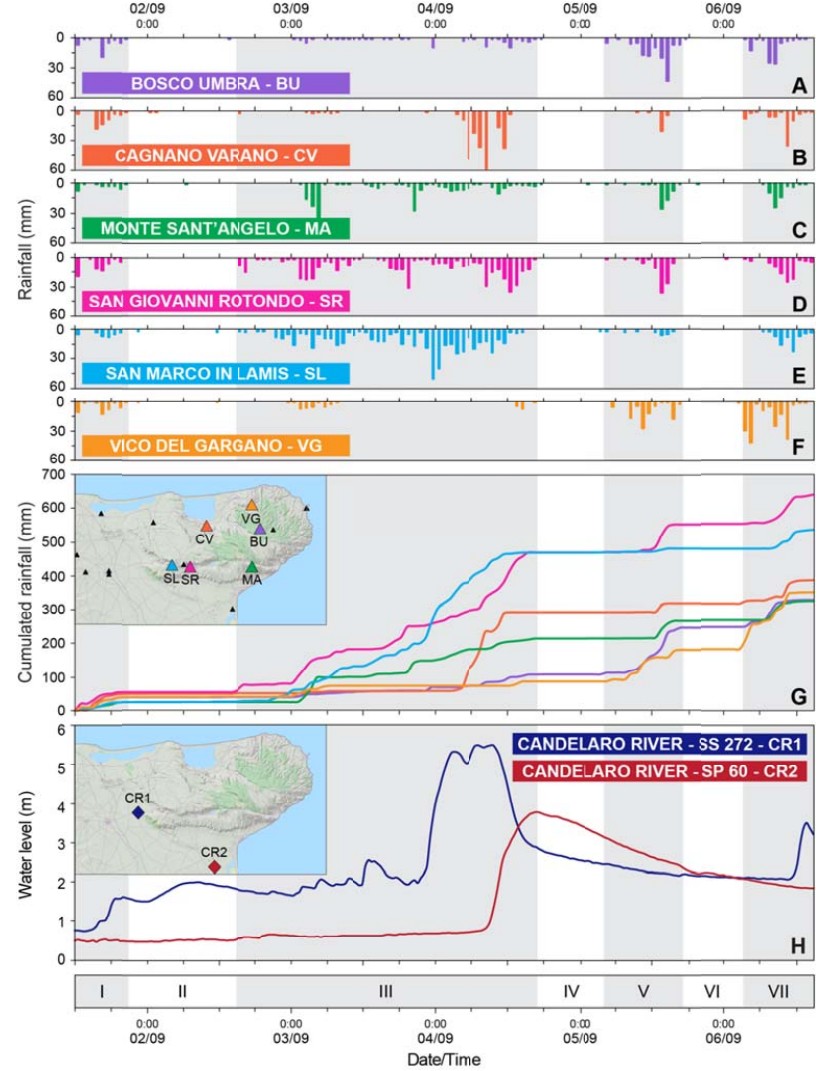

**Figure3: Rainfall and hydrological conditions for the period 1-6 September 2014 in the Gargano Promontory. (A) to (F) hourly rainfall measurements for six rain gauges in the study area. (G) Cumulated rainfall for the same rain gauges; inset shows location of the rain gauges. (H) River water level at two gauging stations along the Candelaro River; inset shows location of gauging stations. In the charts, shaded areas are rainfall ("wet") periods (I, III, V, VII) and white areas are no-rainfall ("dry") periods (II, IV, VI).**




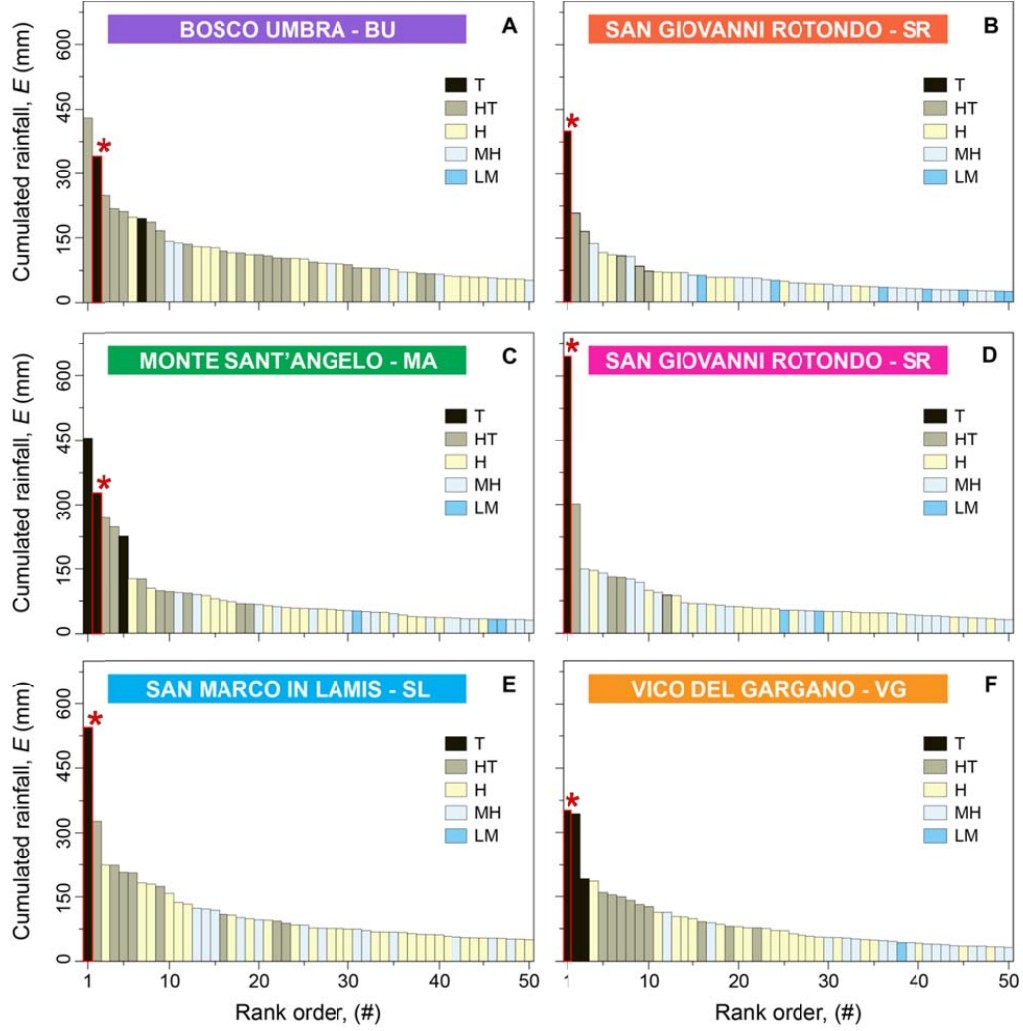

**Figure 4: Rank order analysis of rainfall events in the Gargano Promontory from April 2009 to April 2016. Coloured bars show cumulated event rainfall for six rain gauges: (A) Bosco Umbra - BU, (B) Cagnano Varano - CV, (C) Monte Sant'Angelo - MA, (D) San Giovanni Rotondo - SR, (E) San Marco in Lamis - SM, and (F) Vico del Gargano - VG. Bars arranged from high (left) to low**
5 **(right) values. Colours identify six categories of cumulated rainfall proposed by Alpert et al. (2002). Legend: T, torrential rainfall [128-up mm day$^{-1}$]; HT, Heavy-Torrential [64–128 mm day$^{-1}$]; H, Heavy [32–64 mm day$^{-1}$]; MH, Moderate-Heavy [16–32 mm day$^{-1}$], LM, Light-Moderate [4–16 mm day$^{-1}$]. Black bars with red asterisks show the 1–6 September 2014 period.**




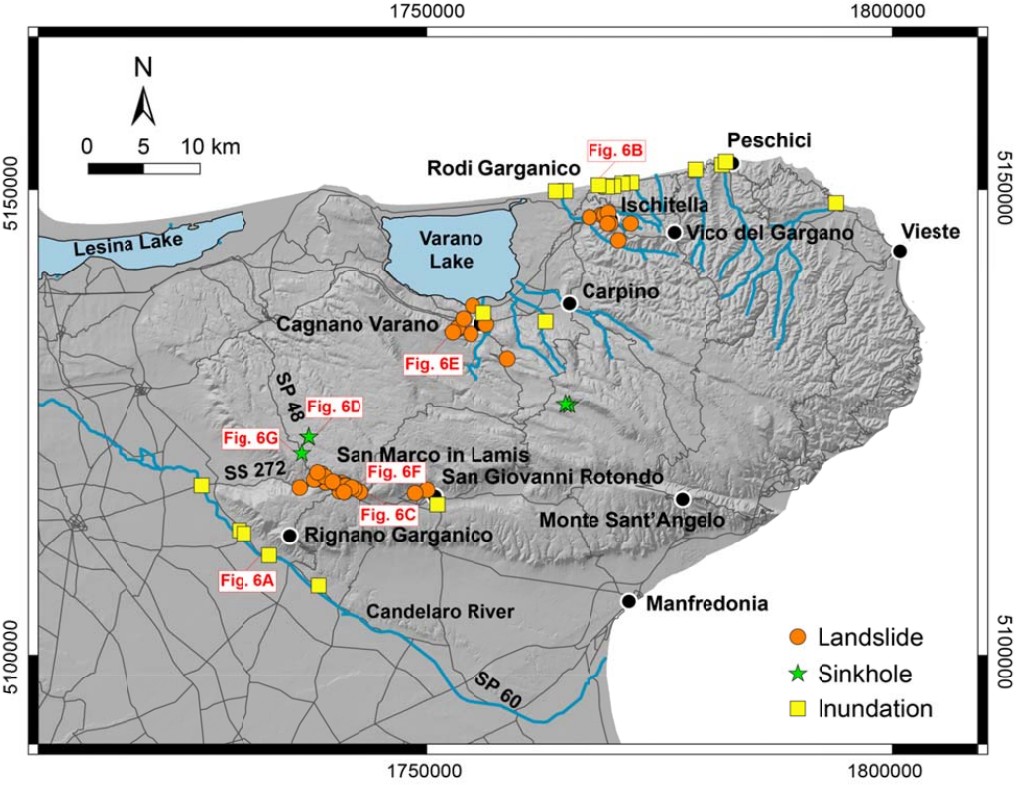

**Figure 5: Map showing location of event landslides, floods, and sinkholes triggered by the 1-6 September 2014, intense rainfall event in the Gargano Promontory. WGS84/Pseudo Mercator (EPSG:3857).**




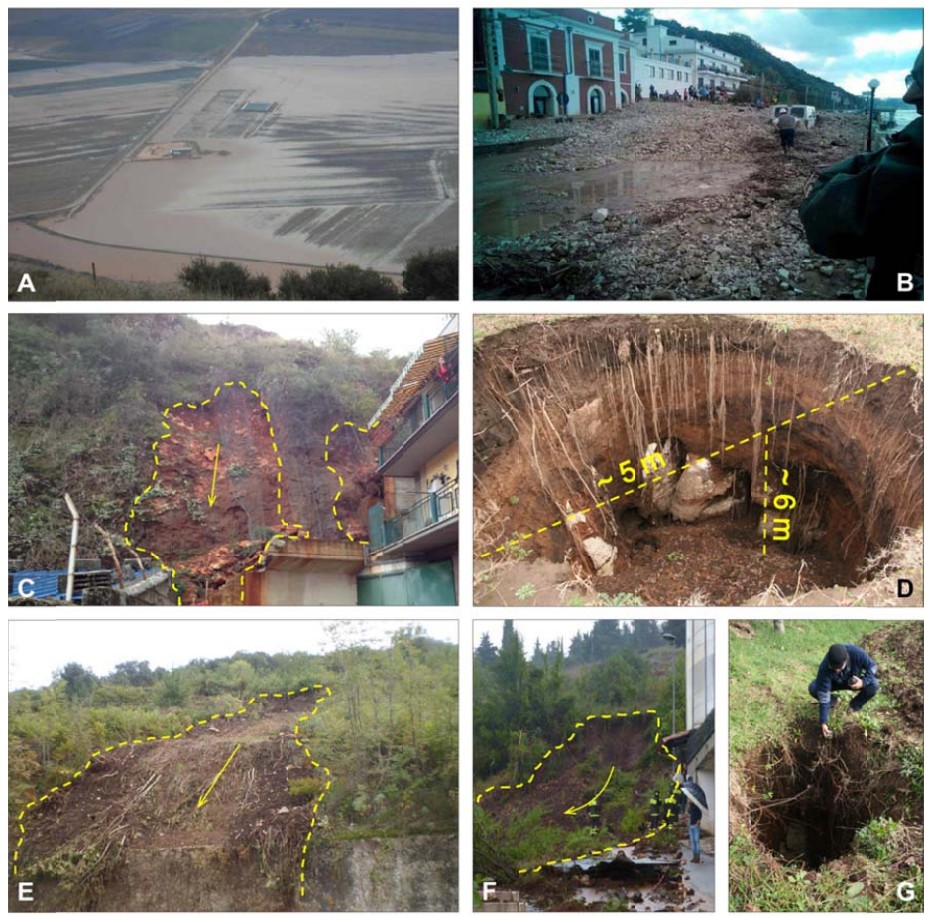

**Figure 6: Examples of geo-hydrological hazards triggered by the 11–6 September 2014 torrential rainfall in the Gargano Promontory. (A) Flood plain inundated by the Candelaro River (photograph: Regione Puglia). (B) Inundation in Rodi Garganico (photograph: Regione Puglia). (C and F) Shallow landslides in the town of San Marco in Lamis. (D) Sinkhole in San Marco in Lamis (photograph: M. Parise). (E) Shallow landslide in Cagnano Varano (photograph: Regione Puglia). (G) Sinkhole in Monte Sant'Angelo (photograph: M. Parise).**


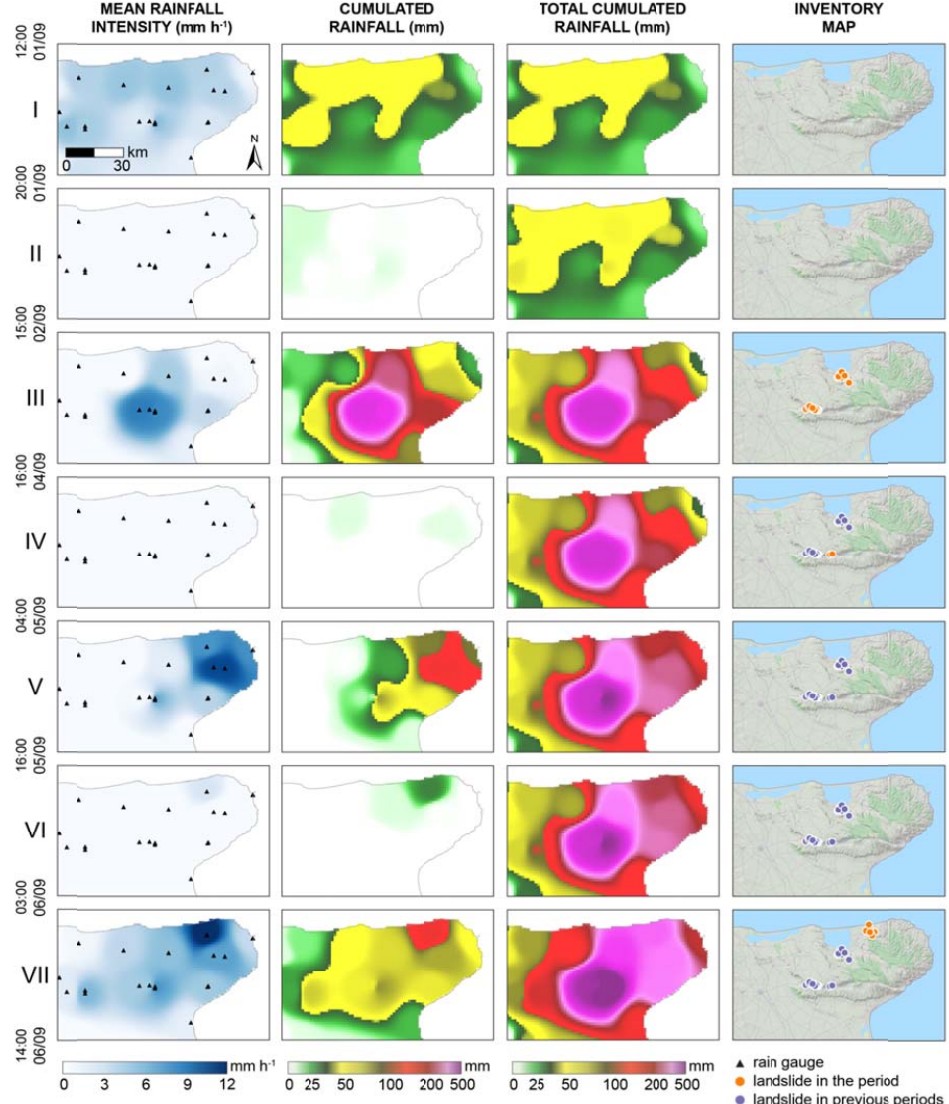

**Figure 7:** Analysis of the spatial and temporal distribution of the event rainfall, and of the triggered event landslides. See text for explanation.




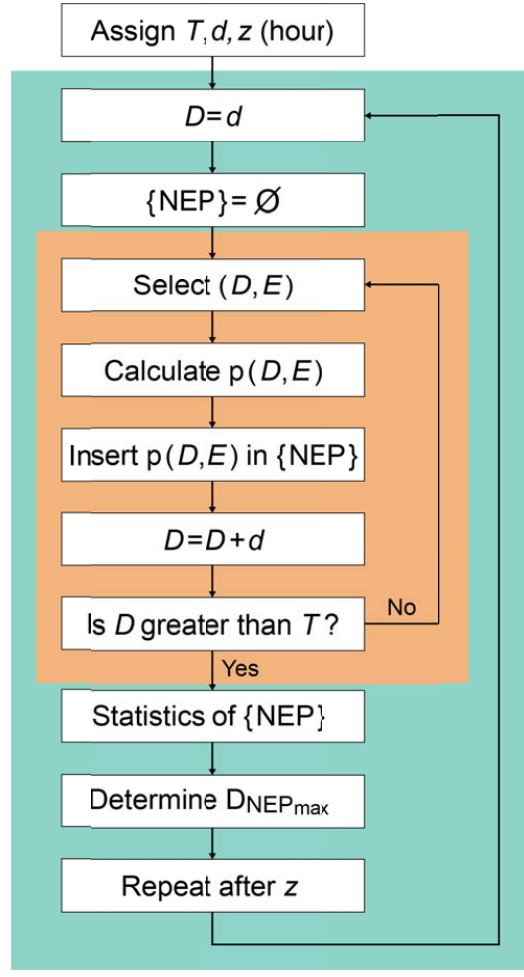

**Figure 8: Logical schema for the E-NEP algorithm.** $D$, rainfall duration, in hours. $E$, cumulated rainfall, in mm. $T$, maximum length of the considered rainfall period, in hours. $d$, time step used to increment the duration of the rainfall period, up to $T$, in hours. $z$, time interval before the next set of $(D,E)$ pairs is computed, in hours. {NEP}, set of non–exceedance probability (NEP) values obtained for each $(D,E)$ pair adopting the method proposed by Brunetti et al. (2010) and Peruccacci et al. (2012). Statistics of {NEP} are 10th, 25th, 50th, 75th, and 90th percentiles. and $NEP_{max}$ is the maximum value of NEP. $D_{NEPmax}$, event rainfall duration corresponding to $NEP_{max}$. External (blue) loop is run every $z$ hours, or fraction of hour. Internal (orange) loop runs from $d$ to $T$, in $d$ time steps. In Figs. 9, 10 $z$ and $d$ were set to 1 hour and $T$ to 96 hours. See text for explanation.



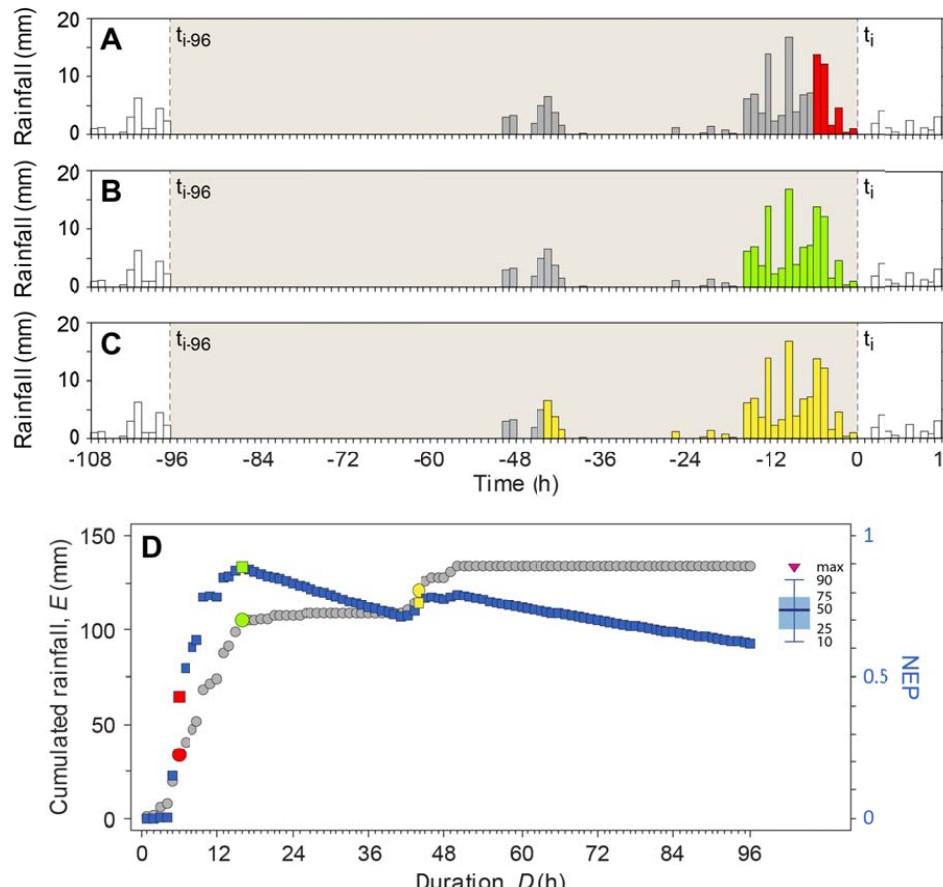

**Figure 9: Exemplification of the E-NEP algorithm used to provide a non-exceedance probability (NEP) of possible landslide occurrence. (A), (B), (C) show the same rainfall record, and three antecedent conditions corresponding to durations of (A) $D$ = 6 hours (red bars), (B) $D$ = 16 hours (green bars), and (C) $D$ = 44 hours (yellow bars). (D) Shows rainfall ($D$,$E$) pairs (grey dots); red, green yellow dots represent the ($D$,$E$) pairs corresponding to the conditions shown in (A), (B), and (C). Blue squares show the corresponding non-exceedance probabilities (NEP). Green square represents NEP$_{max}$. See text for explanation.**



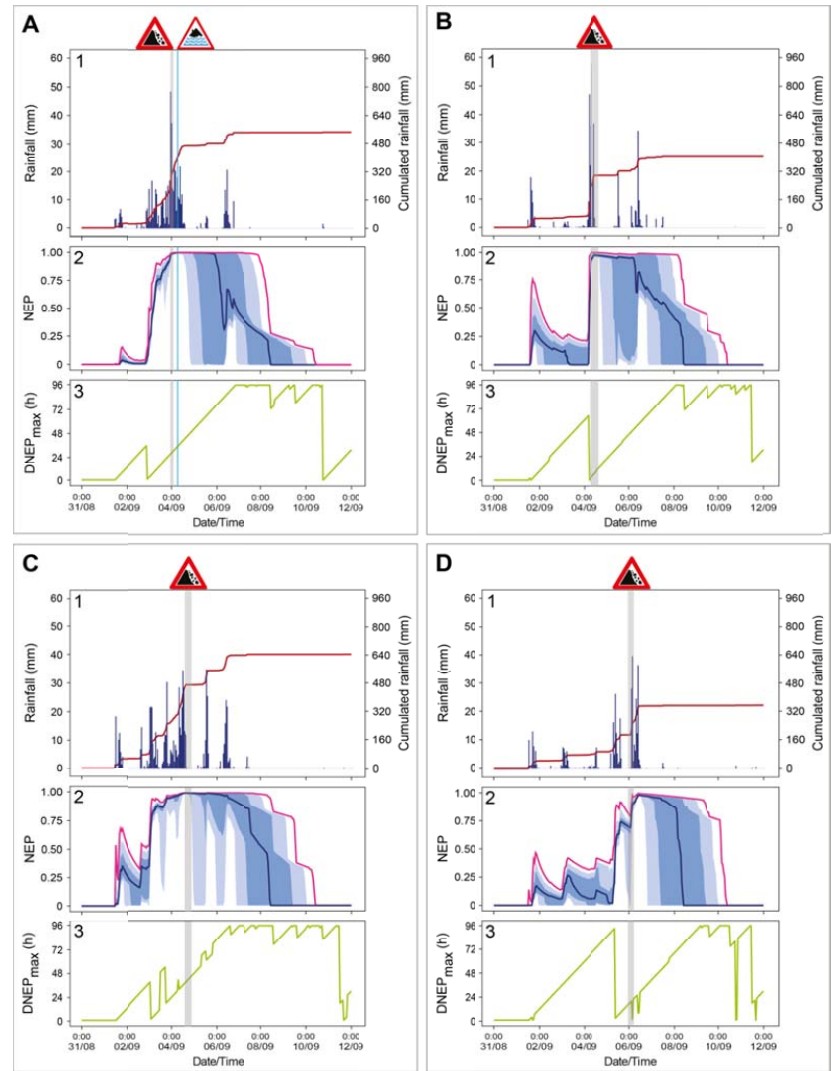

**Figure 10: Results of the application of the E-NEP algorithm to synthetic rainfall records reconstructed for the locations of four landslide clusters. (A) landslide cluster A; (B) landslide cluster B; (C) landslide cluster C; (D) landslide cluster D. Each panel shows, from top to bottom: (1) hourly rainfall record (blue bars) and cumulated event rainfall (red line); (2) median, E-NEP$_{50}$ (orange line) and maximum, E-NEP$_{max}$ (purple line) values of the non-exceedance probability; ranges E-NEP$_{25}$ – E-NEP$_{75}$ (dark blue shade) and E-NEP$_{10}$ – E-NEP$_{90}$ (light blue shade); (3) rainfall duration corresponding to the most-critical rainfall condition, D-NEP$_{max}$ (green line). The period of occurrence of the landslides (identified by the landslide road sign) is shown by a grey shaded area. The time of occurrence of peak flow (identified by the flood road sign) is shown by the vertical blue line. See text for explanation.**