# Peer review of "Landslides, floods and sinkholes in a karst environment: the 1-6 September 2014 Gargano event, southern Italy"

_Natural Hazards and Earth System Sciences, 2016_

## Referee Comment (RC1) · Anonymous Referee #1 · 29 Nov 2016

Review of the manuscript "Landslides, floods and sinkholes in a karst environment: the 1-6 September 2014 Gargano event, southern Italy"

By Maria Elena Martinotti, Luca Pisano, Ivan Marchesini, Mauro Rossi, Silvia Peruccacci, Maria Teresa Brunetti, Massimo Melillo, Giuseppe Amoruso, Pierluigi Loiacono, Carmela Vennari, Giovanna Vessia, Maria Trabace, Mario Parise, Fausto Guzzetti

General Comments The manuscript of Martinotti and co-authors entitled "Landslides, floods and sinkholes in a karst environment: the 1-6 September 2014 Gargano event, southern Italy" is an interesting well-structured and well-written manuscript that addresses relevant scientific and technical questions which are within the scope of NHESS. The authors start with the monography of the hazardous event occurred in

the Gargano Region in September 2014 and finish with the proposition of an algorithm to forecast geo-hydrological hazards. However it needs minor to moderate revisions prior to be published.

Specific Comments

1 – Authors made a very detailed analysis (1 hour time step) of the rainfall event occurred in Gargano region in September 2014, including the evaluation of the Non-Exceedance Probability, and the corresponding duration in hours. However, from the 46 inventoried landslides triggered during the event, the reasonably accurate information about the period of occurrence is only available for 9 landslides. This is a major limitation of the work. Apparently, there is discordance between the detail of the rainfall data and the landslide information.

2 – Taking into consideration comment #1, did authors consider treating the rainfall data with less detail (e.g. cumulative rainfall for each 3 or 6 hours) ?

3 – Can authors provide any information about the exceptionality of the September 2014 rainfall event? What is the estimated return period of the event?

4- Apparently, authors follow the Cruden and Varnes (1996) classification of landslides. However, it is not clear how they distinguish soil slips and soil slides, referred in page 5, line 13.

5 – Referred Landslide clusters A,B,C and D should be clearly showed, for example in figure 5.

6 – Authors verified that significant similarities exist in the temporal evolution of the metrics computed by the E-NEP algorithm. But, to which extent is this results controlled by the specific characteristics of the registered rainfall events?

7 - Looking on the period VI in Table 1, it is arguable to consider as "dry" a 11-hour period with 1.8 mm/hour rainfall intensity.

Technical corrections

Page 4 line 23 "record available" instead of "record available to us".

Page 4 line 25 Include a reference to figure 5 (e.g. after sinkholes). It is desirable that figure 5 s referred in text prior to figure 6.

Page 5 line 7 "the towns of Cagnano Varano and Carpino (Fig. 5)".

Page 8 line 14 Reference to figure 2 is inadequate.

Page 8 line 17 Reference to figure 3 is inadequate.

Page 10 line 25 The reviewer would not include rock falls in the group of landslides authors are dealing with.

Page 11 line 14 "High infiltration to shallow depth in the rock mass facilitates the formation of flash floods". This is not clear.

Figure 2 Reconsider the caption of figure 2. The area covered by images is much larger than the Gargano Promontory. I suggest using "over the central and southern Italy" instead of "in the Gargano Promontory".

Figure 6 The caption of photo F is missing. 1-6 September instead of 11-6 September.

Figure 7 Landslide inventory map instead of Inventory map.

---

## Referee Comment (RC2) · J. De Waele (Referee) · 29 Nov 2016

General comments The paper is well constructed, and describes an exceptional rainfall event and the natural hazards it created. It analyses rainfall data accurately and relates this event to the hazards that occurred in the territory using the E-NEP algorithm. It fails however to validate this new evaluation method (as early warning system) because of a scarce knowledge on the timing of hazardous events. This paper thus does not really give a great advance in what we know about natural hazards caused by heavy rainfall. It simply describes an exceptional rain event in detail, gives some general information on the hazards this event caused, and then explains that these happened because of heavy rainfall and high intensity. Nothing really new. The method looks promising

(E-NEP) but would need to be validated on a more detailed database of well-know hazards of which the timing is known in detail.

Detailed comments 1. you overuse many terms, and parts of sentences. Why geo-hydrological hazards? I would simply use hydrogeological hazards. The word (geo-hydrological) is used too often throughout the manuscript (6 times in the introduction only!). Avoid repeating too much terms like this, or the list of hazards (landslides, flash floods, inundations and sinkholes).

2. There is not much geological information on the landslides or other hazards. Of course they cannot be explained all in detail, but a table should be included giving some information on each landslide, sinkhole, flood... material involved, type of landslide, thickness of soil, inclination, dimensions, exposition, altitude, soil cover... Landslide are caused by many factors, not only rain. You simplify also a lot. E.g. Soils are thin or absent... looking at Figure 6 I see a lot of soil!

3. Figure 5: I would show the clusters (A, B, ...) on this map, and also use smaller symbols for the registered events (now I count 30 circles for landslides, while there should be 46!). I also suggest to distinguish the types with different symbols or colors. If needed the three most densely packed areas might be increased in size (use insets and detailed maps if needed). What are the difference between soils slips and soil slides? Did all these events REALLY occur during the rainfall events (in this period) or are some older or occurred after the considered period. How much fieldwork has been done here? It almost looks as if most of the work has been done on a computer without really studying the hazards in the field. Did you mostly work on a database given by different authorities (like civil protection, fire corps, forestry department, mairs,...).

4. Self-citation: Parise is cited 8 out of 28, Guzzetti 4 etc. Some citation are probably not really needed (Cannon et al. 2003, De Graff et al, 2013)

5. It is strange to put different hazards together. Sinkholes, floods and landslides do not have a lot of genetic mechanisms in common. Of course they form more easily under

heavy rainfall, but that is their only important common point. I would have focussed on landslides, trying to have more detail on those...

Minor adjustments (I used line numbers) Page 1 15 hydrogeological hazards (I suggest changing this throughout the text) 17 a karst area in Apulia 19 and temporal information 24 (E-NEP) Page 2 6 delete "by landslides.... inundations" 11 characteristics ... delete "Landslides.... sinkholes" and write "natural hazards" instead 20 Apulia (delete Puglia (...)) 26 In the area sedimentary rocks crop out... Page 3 2 have been reported (instead of exist)... delete "including....sinkholes" 11 metres 13 Description of events 26 stations (instead of rain gauges) 30 dry periods lasting between Page 4 8 in this period 25 delete "landslides and sinkholes" 35 translate water height (5.30 m) in flow rate. Page 5 1 and 3 flow rate instead of water height 2 from (not form) 6 Cagnano-Carpino was a landslide fatality, the flood fatality was at Peschici (see line 25 on page 4). Make up your mind! 12 mostly shallow landslides 13 write 4 and 2 instead of four and two 22 delete "of landslides" Page 6 17 of cumulated 18 of rainfall ..... exceeded... Page 7 1-3 you use 3 times geo-hydrological! use hydrogeological (possibly) and delete sometimes (just use hazards) 6 Probablity (E-NEP) algorithm ... record (delete s) 10 and d the time... 18 delete "landslides..... hazards" 28 delete else (first word) and write otherwise Page 8 2 possible hazard occurrence ... was calculated (delete using the approach.... (2012) Page 9 6 The analysis ...... DNEPmax is of interest 22 rise following 27 (E-NEP) 32 I, on their own, were not.... Page 10 15 et al., 2007 30 delete 2003 (not needed) and De Graff et al., 2013 These are difficult-to-find papers. 32 delete "that lays.... E-NEP." Page 11 2 delete (Brunetti...... 2012) 7 driven hazards 10 these hazards 30 Apulia Page 12 10 (E-NEP) 11 sinkholes). insert space For... 20 events they are abundant 21 as for the Page 14 9 Szonyi Page 17 5 sea level Page 18 I would have placed the rainfall station in a more logical sequence (from N to S? Or from E to W). Now they are placed rather randomly. Page 21 2 1-6 Page 23 2 scheme Page 25 The symbols of landslide are placed in ace rain time interval rather precisely, although you stated that only 9 were known to have occurred at precise intervals. Is this just a graphical representation? Or do you ONLY mention the 9 known ones. Explain please.

---

## Author Comment (AC2) · 23 Jan 2017

We thank the Reviewer for his comments. The scope of the paper is indeed the development of a new algorithm (E-NEP) to analyze rainfall events responsible for geo-hydrological hazards in order to predict their occurrence. The tool was successfully applied to the 2014 Gargano event, where it turned to be able to predict the occurrence of the observed landslides. We maintain that the new algorithm can contribute to forecast the possible occurrence of rainfall-induced landslides and to ascertain landslide hazard. Below, we address detailed comments and describe the modifications we have made to the manuscript.

1 - The text was amended in order to avoid repetitions. However, we prefer to use the

term "geo-hydrological" instead of "hydrogeological". We believe that the term "geo-hydrological hazards" is more apt to delineate the range of phenomena that encompass, among the others, the landslides, floods, debris flows, sinkholes, erosions. In addition to a wide use of the term geo-hydrological in the scientific literature, this is also explained in a recent document discussing the Italian National Strategy for the Adaptation to the Climatic Changes (SNAC): Castellari S., Venturini S., Ballarin Denti A., Bigano A., Bindi M., Bosello F., Carrera L., ChiriacoÌĂ M.V., Danovaro R., Desiato F., Filpa A., Gatto M., Gaudioso D., Giovanardi O., Giupponi C., Gualdi S., Guzzetti F., Lapi M., Luise A., Marino G., Mysiak J., Montanari A., Ricchiuti A., Rudari R., Sabbioni C., Sciortino M., Sinisi L., Valentini R., Viaroli P., Vurro M., Zavatarelli M. (a cura di.) (2014). Rapporto sullo stato delle conoscenze scientifiche su impatti, vulnerabilitaÌĂ ed adattamento ai cambiamenti climatici in Italia. Ministero dell'Ambiente e della Tutela del Territorio e del Mare, Roma.

2 - We acknowledge that geological and geomorphological information on the hazards is quite poor, but it is not relevant for the application of the E-NEP algorithm. Nevertheless, we modified section 3.3 adding more details about the data, as follows: "The consequences of the storm of September 2014 were reported soon after their occurrence, and a first analysis was carried out immediately in its aftermath. The collection of information was obtained searching different sources: (i) field surveys, (ii) technical reports produced by geologists; and (iii) on-line national, regional, and local newspapers. The collected information allowed reconstructing the geographical coordinates of each phenomenon, its occurrence date, and the type of hazard. No geological and geomorphological details were available for the landslides, especially when the information was found in newspapers. A specific landslide catalogue was built and managed in a GIS environment. The catalogue lists the following items: (i) event identification code, (ii) source of information, (iii) landslide location (geographic coordinates, municipality, province), (iv) occurrence date and time (if available), (v) spatial and temporal accuracy, and (vi) landslide type. As concerns floods, the main information regarded the interested areas, the reported damage, and the extent of the flooded territory. Information on sinkholes included the occurrence site obtained through field surveys (high geographical accuracy), and the occurrence time, which was mostly based upon interviews with local inhabitants (low to medium temporal accuracy)." We acknowledge that the statement about the soil thickness is too much generic and we removed it from the manuscript.

3 - We thank the Reviewer for advices. We confirm that the map contains 46 circles, but some of them are really close to each other or overlapped. Smaller symbols would make them not visible. According to the Reviewer suggestion, we used different symbols and colors to describe the different types of hazards. We think that using insets would make the figure too much complex without providing additional and substantial information. We acknowledge that there was an error and soil slips of the original manuscript must be intended as earth flows. We have amended the text accordingly. In the reply to comment 2 we specified that: "The consequences of the storm of September 2014 were reported soon after their occurrence, and a first analysis was carried out immediately in its aftermath. The collection of information was obtained searching different sources: (i) field surveys, (ii) technical reports produced by geologists; and (iii) on-line national, regional, and local newspapers".

4 - Some of the self-citation have been deleted, in order the reduce their overall number. However, we have to point out that there are not many works about hazards in karst in the study area, which forced in some ways to cite the existing ones, most of which include authors of the present manuscript.

5 - The paper deals with the natural hazards triggered by the September 2014 storm. Two outcomes of the manuscript are that: (i) landslides are not the only hazards that may occur in the Gargano promontory during heavy rainfalls and (ii) more accurate information are needed on the time or period of occurrence of sinkholes. Indeed, we cannot exclude that even those hazard, analogously to landslides and flash floods, can be likely predicted using rainfall thresholds. This was reported at page 11 of the original manuscript: "Based on the analysis of 25 the 1-6 September 2014 Gargano

rainfall period, we confirm that in the Promontory, and in similar karst areas, torrential rainfall can trigger sinkholes, and we hypothesise that approaches based on the near-real-time monitoring of rainfall (e.g., the E-NEP algorithm) can be used to forecast the possible occurrence of rainfall-induced sinkholes. We acknowledge that an analysis of a larger number of events is required to test this hypothesis.".

Minor adjustments Page 1 15 hydrogeological hazards (I suggest changing this throughout the text). We removed other occurrences of the term "geo-hydrological".

Page 1 17 a karst area in Apulia. We would prefer using the Italian name Puglia in place of Apulia. The correspondence with the international name Apulia is now given in section 1, where we write "Puglia (Apulia)".

Page 1 19 and temporal information. Done.

Page 1 24 (E-NEP). Done.

Page 2 6 delete "by landslides.... inundations". Done.

Page 2 11 characteristics ... delete "Landslides.... sinkholes" and write "natural hazards" instead. We prefer to list the types of natural hazards discussed in the paper.

Page 2 20 Apulia (delete Puglia (...)) We would prefer using the Italian name Puglia in place of Apulia.

Page 2 26 In the area sedimentary rocks crop out... Done.

Page 3 2 have been reported (instead of exist)... delete "including....sinkholes". Done.

Page 3 11 metres. Done.

Page 3 13 Description of events. Done.

Page 3 26 stations (instead of rain gauges). We decided to maintain "rain gauges".

Page 3 30 dry periods lasting between. Done.

Page 4 8 in this period. Done.

Page 4 25 delete "landslides and sinkholes". We rephrased as follows: "in a number of floods, flash floods, landslides and sinkholes . . .".

Page 4 35 translate water height (5.30 m) in flow rate. We do not have the stage-discharge relationship for the cross-section and so it is not possible to transform the height in flow rate.

Page 5 1 and. Done.

Page 5 3 flow rate instead of water height. As above.

Page 5 2 from (not form). Done.

Page 5 6 Cagnano-Carpino was a landslide fatality, the flood fatality was at Peschici (see line 25 on page 4). Make up your mind! We thank the Reviewer. Actually, there was an error in the data we received. Both the fatalities were due to floods. We amended the text consequently.

Page 5 12 mostly shallow landslides. Done.

Page 5 13 write 4 and 2 instead of four and two. Done.

Page 5 22 delete "of landslides". Done.

Page 6 17 of cumulated. Done.

Page 6 18 of rainfall ..... exceeded... Done.

Page 7 1-3 you use 3 times geo-hydrological! use hydrogeological (possibly) and delete sometimes (just use hazards). Done.

Page 7 6 Probability (E-NEP) algorithm ... record (delete s). Done.

Page 7 10 and d the time... Done.

Page 7 18 delete "landslides..... hazards". We decided to keep it.

Page 7 28 delete else (first word) and write otherwise. Done.

Page 8 2 possible hazard occurrence ... was calculated (delete using the approach.... (2012). Done.

Page 9 6 The analysis ...... DNEPmax is of interest. Done.

Page 9 22 rise following. Done.

Page 9 27 (E-NEP). Done.

Page 9 32 I, on their own, were not.... Done.

Page 10 15 et al., 2007. Done.

Page 10 30 delete 2003 (not needed) and De Graff et al., 2013 These are difficult-to-find papers. Done.

Page 10 32 delete "that lays.... E-NEP." Done.

Page 11 2 delete (Brunetti...... 2012). Done.

Page 11 7 driven hazards. Done.

Page 11 10 these hazards. Done.

Page 11 30 Apulia. As above.

Page 12 10 (E-NEP). Done.

Page 12 11 sinkholes). insert space For... Done.

Page 12 20 events they are abundant. Done.

Page 12 21 as for the. Done.

Page 14 9 Szonyi. Done.

Page 17 5 sea level. Done.

Page 18 I would have placed the rainfall station in a more logical sequence (from N to S? Or from E to W). Now they are placed rather randomly. Done, now they are placed N to S.

Page 21 2 1-6. Done.

Page 23 2 scheme. Done.

Page 25 The symbols of landslide are placed in ace rain time interval rather precisely, although you stated that only 9 were known to have occurred at precise intervals. Is this just a graphical representation? Or do you ONLY mention the 9 known ones. Explain please. We have temporal information only about 9 landslides (4 in cluster A, 1 in cluster B, 1 in cluster C, 3 in cluster D) but we assume these landslides to be representative of the clusters (this is explained in Section 3.3). The landslide occurrence time is provided with an estimated uncertainty represented by the shadowed gray band in Figure 10. We modified the caption to explain it better. In particular, we wrote: "The occurrence time (and the associated uncertainty) of nine landslides (cfr. section 3.3) is used to define the landslide occurrence period of the four clusters . . .".
* * *
**Fig. 1.** Figure3: Rainfall and hydrological conditions for the period 1-6 September 2014 in the Gargano Promontory. (A) to (F) hourly rainfall measurements for six rain gauges in the study area. (G) Cumulated

**Fig. 2.** Figure 5: Map showing location of event landslides, floods, and sinkholes triggered by the 1-6 September 2014, intense rainfall event in the Gargano Promontory. WGS84/Pseudo Mercator (EPSG:3857).

---

## Author Response (AR2)

Reply to the reviewer comments for the Martinotti et al. 2016 manuscript "Landslides, floods and sinkholes in a karst environment: the 1-6 September 2014 Gargano event, southern Italy"

**Reviewer 1: Anonymous**

The manuscript of Martinotti and co-authors entitled "Landslides, floods and sinkholes in a karst environment: the 1-6 September 2014 Gargano event, southern Italy" is an interesting well-structured and well-written manuscript that addresses relevant scientific and technical questions which are within the scope of NHESS. The authors start with the monography of the hazardous event occurred in the Gargano Region in September 2014 and finish with the proposition of an algorithm to forecast geo-hydrological hazards. However it needs minor to moderate revisions prior to be published.

*We thank the Reviewer for considering the manuscript interesting, well-written and well-structured. We are particularly thankful for her/his opinion about the fact that the manuscript addresses relevant scientific and technical questions.*

**Specific Comments**

1. Authors made a very detailed analysis (1 hour time step) of the rainfall event occurred in Gargano region in September 2014, including the evaluation of the NonExceedance Probability, and the corresponding duration in hours. However, from the 46 inventoried landslides triggered during the event, the reasonably accurate information about the period of occurrence is only available for 9 landslides. This is a major limitation of the work. Apparently, there is discordance between the detail of the rainfall data and the landslide information.

*The time resolution of rainfall data and the temporal accuracy of the landslides are independent. We attribute the poor accuracy in the occurrence time of the majority of the failures to the difficulty to reach some of the places where the landslides occurred, and to the fact that many landslides occurred in the evening or during the night, and were reported only several hours after the event. Nevertheless, we want to stress that even if the temporal uncertainty is large, plots in Figure 10 show that the temporal ranges defined for the possible occurrence of the landslides do not significantly affect the prediction capabilities of the E-NEP algorithm.*

2. Taking into consideration comment #1, did authors consider treating the rainfall data with less detail (e.g. cumulative rainfall for each 3 or 6 hours)?

*We did not take into consideration the hypothesis of considering different intervals for the analysis of the cumulative rainfalls. However, the outcome of such analysis would have resulted in less detailed plots (e.g. cfr. Fig. 10) with histograms having larger bins and lines showing abrupt steps. We think that we would not have obtained relevant information from the observations of those plots.*

3. Can authors provide any information about the exceptionality of the September 2014 rainfall event? What is the estimated return period of the event?

*Unfortunately, we only have rainfall data for a 7-years period for the selected rain gauges. For this reason, we cannot add nothing more than what we have reported in the manuscript (Figure 4): except for the Monte Sant'Angelo rain gauge, the 1–6 September rainfall period exhibited the highest cumulated rainfall in the observation period.*

4. Apparently, authors follow the Cruden and Varnes (1996) classification of landslides. However, it is not clear how they distinguish soil slips and soil slides, referred in page 5, line 13.

*There was an error in preparing the first version of the manuscript, and the soil slips are actually earth flows. We amended the text accordingly.*

5. Referred Landslide clusters A,B,C and D should be clearly showed, for example in figure 5.

*We thank the Reviewer for the suggestion that improves the readability of the paper and of the figure. Figure 5 now includes the location of the clusters.*

6. Authors verified that significant similarities exist in the temporal evolution of the metrics computed by the E-NEP algorithm. But, to which extent is this results controlled by the specific characteristics of the registered rainfall events?

*This is a good point and we are thankful to the Reviewer for the question. As stated in section 5 (Discussion), the analysis of the rainfall records (Fig. 10) highlights that neither the rainfall intensity nor the cumulative rainfall could be considered diagnostic for the detection of the rainfall conditions responsible for landslides. We also observe that the evolution of the precipitation, before and during the landslide occurrence time, was different among the four clusters. In the different clusters, the specific characteristics of the rainfall events are different. Surprisingly, the E-NEP metric, in the period of the landslide occurrence, is not affected by this variability. Landslides have occurred, in all the cases, as soon as (i) the $NEP_{max}$ metric reached the maximum value and (ii) all the NEP percentiles increase abruptly. We conclude that, at least for this study, the results obtained using the E-NEP metrics are not controlled by the specific characteristics of the rainfall events.*

7. Looking on the period VI in Table 1, it is arguable to consider as "dry" a 11-hour period with 1.8 mm/hour rainfall intensity.

*Thank you for the remark. We used the term nearly-dry for classifying that period in the manuscript.*

**Minor adjustments**

line 23 "record available" instead of "record available to us".
*Done.*

line 25 Include a reference to figure 5 (e.g. after sinkholes). It is desirable that figure 5 s referred in text prior to figure 6.

*We thank the Reviewer for her/his advice, we added a citation of Figure 5 before that of Figure 6.*

**Page 5**

line 7 "the towns of Cagnano Varano and Carpino (Fig. 5)".
*Done.*

**Page 8**

line 14 Reference to figure 2 is inadequate.
*Done. Reference deleted.*

line 17 Reference to figure 3 is inadequate.
*Done. Reference deleted.*

**Page 10**

line 25 The reviewer would not include rock falls in the group of landslides authors are dealing with.
*It is true that rock falls are the landslides types that are less related to rainfall. We removed them from the text.*

**Page 11**

line 14 "High infiltration to shallow depth in the rock mass facilitates the formation of flash floods". This is not clear.
*In order to better explain the meaning of the previous sentence, the paragraph has been modified and rewritten as follows.*

*"In the karst environment of the Promontory, rainfall infiltration is efficient even for high intensity rainfall rates. This limits the occurrence of landslides, except for very intense (i.e., "extreme") rainfall events. On the other hand, arrival of great amount of rainfall in a setting typically characterized by water infiltrating within the rock mass through the network of conduits and joints, highly facilitates the formation of flash floods, particularly in small catchments, as has been frequently registered also in other parts of Puglia (Parise, 2003; Mossa, 2007). Further, karst aquifers have very poor retention capacity. These, and other characteristics as well, allow to identify the flash floods as one of the main hazards in karst terrains (Fleury et al., 2013; Gutierrez et al., 2014; Parise et al., 2015)."*

**Figure 2**

Reconsider the caption of figure 2. The area covered by images is much larger than the Gargano Promontory. I suggest using "over the central and southern Italy" instead of "in the Gargano Promontory".
*Done.*

**Figure 6**

The caption of photo F is missing. 1-6 September instead of 11-6 September.
*Photo F is considered together with photo C in the figure caption. We corrected the dates.*

**Figure 7**

Landslide inventory map instead of Inventory map.
*Done.*

**Reviewer 2: J. De Waele (jo.dewaele@unibo.it)**

**General comments**

The paper is well constructed, and describes an exceptional rainfall event and the natural hazards it created. It analyses rainfall data accurately and relates this event to the hazards that occurred in the territory using the E-NEP algorithm. It fails however to validate this new evaluation method (as early warning system) because of a scarce knowledge on the timing of hazardous events. This paper thus does not really give a great advance in what we know about natural hazards caused by heavy rainfall. It simply describes an exceptional rain event in detail, gives some general information on the hazards this event caused, and then explains that these happened because of heavy rainfall and high intensity. Nothing really new. The method looks promising (E-NEP) but would need to be validated on a more detailed database of well-known hazards of which the timing is known in detail.

*We thank the Reviewer for his comments. The scope of the paper is indeed the development of a new algorithm (E-NEP) to analyze rainfall events responsible for geo-hydrological hazards in order to predict their occurrence. The tool was successfully applied to the 2014 Gargano event, where it turned to be able to predict the occurrence of the observed landslides. We maintain that the new algorithm can contribute to forecast the possible occurrence of rainfall-induced landslides and to ascertain landslide hazard.*

*Below, we address detailed comments and describe the modifications we have made to the manuscript.*

**Detailed comments**

1. you overuse many terms, and parts of sentences. Why geohydrological hazards? I would simply use hydrogeological hazards. The word (geohydrological) is used too often throughout the manuscript (6 times in the introduction only!). Avoid repeating too much terms like this, or the list of hazards (landslides, flash floods, inundations and sinkholes).

*The text was amended in order to avoid repetitions. However, we prefer to use the term "geo-hydrological" instead of "hydrogeological". We believe that the term "geo-hydrological hazards" is more apt to delineate the range of phenomena that encompass, among the others, the landslides, floods, debris flows, sinkholes, erosions. In addition to a wide use of the term geo-hydrological in the scientific literature, this is also explained in a recent document discussing the Italian National Strategy for the Adaptation to the Climatic Changes (SNAC): Castellari S., Venturini S., Ballarin Denti A., Bigano A., Bindi M., Bosello F., Carrera L., Chiriacò M.V., Danovaro R., Desiato F., Filpa A., Gatto M., Gaudioso D., Giovanardi O., Giupponi C., Gualdi S., Guzzetti F., Lapi M., Luise A., Marino G., Mysiak J., Montanari A., Ricchiuti A., Rudari R., Sabbioni C., Sciortino M., Sinisi L., Valentini R., Viaroli P., Vurro M., Zavatarelli M. (a cura di.) (2014). Rapporto sullo stato delle conoscenze scientifiche su impatti, vulnerabilità ed adattamento ai cambiamenti climatici in Italia. Ministero dell'Ambiente e della Tutela del Territorio e del Mare, Roma.*

2. There is not much geological information on the landslides or other hazards. Of course they cannot be explained all in detail, but a table should be included giving some information on each landslide, sinkhole, flood... material involved, type of landslide,

thickness of soil, inclination, dimensions, exposition, altitude, soil cover... Landslide are caused by many factors, not only rain. You simplify also a lot. E.g. Soils are thin or absent... looking at Figure 6 I see a lot of soil!

*We acknowledge that geological and geomorphological information on the hazards is quite poor, but it is not relevant for the application of the E-NEP algorithm.*

*Nevertheless, we modified section 3.3 adding more details about the data, as follows: "The consequences of the storm of September 2014 were reported soon after their occurrence, and a first analysis was carried out immediately in its aftermath. The collection of information was obtained searching different sources: (i) field surveys, (ii) technical reports produced by geologists; and (iii) on-line national, regional, and local newspapers.*

*The collected information allowed reconstructing the geographical coordinates of each phenomenon, its occurrence date, and the type of hazard.*

*No geological and geomorphological details were available for the landslides, especially when the information was found in newspapers. A specific landslide catalogue was built and managed in a GIS environment. The catalogue lists the following items: (i) event identification code, (ii) source of information, (iii) landslide location (geographic coordinates, municipality, province), (iv) occurrence date and time (if available), (v) spatial and temporal accuracy, and (vi) landslide type.*

*As concerns floods, the main information regarded the interested areas, the reported damage, and the extent of the flooded territory. Information on sinkholes included the occurrence site obtained through field surveys (high geographical accuracy), and the occurrence time, which was mostly based upon interviews with local inhabitants (low to medium temporal accuracy)."*

*We acknowledge that the statement about the soil thickness is too much generic and we removed it from the manuscript.*

3. Figure 5: I would show the clusters (A, B, ...) on this map, and also use smaller symbols for the registered events (now I count 30 circles for landslides, while there should be 46!). I also suggest to distinguish the types with different symbols or colors. If needed the three most densely packed areas might be increased in size (use insets and detailed maps if needed). What are the difference between soils slips and soil slides? Did all these events REALLY occur during the rainfall events (in this period) or are some older or occurred after the considered period. How much fieldwork has been done here? It almost looks as if most of the work has been done on a computer without really studying the hazards in the field. Did you mostly work on a database given by different authorities (like civil protection, fire corps, forestry department, mairs, ).

*We thank the Reviewer for advices. We confirm that the map contains 46 circles, but some of them are really close to each other or overlapped. Smaller symbols would make them not visible. According to the Reviewer suggestion, we used different symbols and colors to describe the different types of hazards. We think that using insets would make the figure too much complex without providing additional and substantial information.*

*We acknowledge that there was an error and soil slips of the original manuscript must be intended as earth flows. We have amended the text accordingly.*

*In the reply to comment 2 we specified that: "The consequences of the storm of September 2014 were reported soon after their occurrence, and a first analysis was carried out immediately in its aftermath. The collection of information was obtained searching different*

*sources: (i) field surveys, (ii) technical reports produced by geologists; and (iii) on-line national, regional, and local newspapers".*

4. Self-citation: Parise is cited 8 out of 28, Guzzetti 4 etc. Some citations are probably not really needed (Cannon et al. 2003, De Graff et al, 2013).

*Some of the self-citation have been deleted, in order the reduce their overall number. However, we have to point out that there are not many works about hazards in karst in the study area, which forced in some ways to cite the existing ones, most of which include authors of the present manuscript.*

5. It is strange to put different hazards together. Sinkholes, floods and landslides do not have a lot of genetic mechanisms in common. Of course they form more easily under heavy rainfall, but that is their only important common point. I would have focused on landslides, trying to have more detail on those...

*The paper deals with the natural hazards triggered by the September 2014 storm. Two outcomes of the manuscript are that: (i) landslides are not the only hazards that may occur in the Gargano promontory during heavy rainfalls and (ii) more accurate information are needed on the time or period of occurrence of sinkholes. Indeed, we cannot exclude that even those hazard, analogously to landslides and flash floods, can be likely predicted using rainfall thresholds. This was reported at page 11 of the original manuscript: "Based on the analysis of 25 the 1-6 September 2014 Gargano rainfall period, we confirm that in the Promontory, and in similar karst areas, torrential rainfall can trigger sinkholes, and we hypothesise that approaches based on the near-real-time monitoring of rainfall (e.g., the E-NEP algorithm) can be used to forecast the possible occurrence of rainfall-induced sinkholes. We acknowledge that an analysis of a larger number of events is required to test this hypothesis.".*

**Minor adjustments**

15 hydrogeological hazards (I suggest changing this throughout the text).
*We removed other occurrences of the term "geo-hydrological".*

17 a karst area in Apulia.
*We would prefer using the Italian name Puglia in place of Apulia. The correspondence with the international name Apulia is now given in section 1, where we write "Puglia (Apulia)".*

19 and temporal information.
*Done.*

24 (E-NEP)
*Done.*

6 delete "by landslides.... inundations".

*Done.*

11 characteristics ... delete "Landslides.... sinkholes" and write "natural hazards" instead.
*We prefer to list the types of natural hazards discussed in the paper.*

20 Apulia (delete Puglia (...))
*We would prefer using the Italian name Puglia in place of Apulia.*

26 In the area sedimentary rocks crop out...
*Done.*

**Page 3**

2 have been reported (instead of exist)... delete "including....sinkholes"
*Done.*

11 metres
*Done.*

13 Description of events
*Done.*

26 stations (instead of rain gauges)
*We decided to maintain "rain gauges".*

30 dry periods lasting between
*Done.*

**Page 4**

8 in this period
*Done.*

25 delete "landslides and sinkholes"
*We rephrased as follows: "in a number of floods, flash floods, landslides and sinkholes …".*

35 translate water height (5.30 m) in flow rate.
*We do not have the stage-discharge relationship for the cross-section and so it is not possible to transform the height in flow rate.*

**Page 5**

1 and
*Done.*

3 flow rate instead of water height
*As above.*

2 from (not form)
*Done.*

6 Cagnano-Carpino was a landslide fatality, the flood fatality was at Peschici (see line 25 on page 4). Make up your mind!

*We thank the Reviewer. Actually, there was an error in the data we received. Both the fatalities were due to floods. We amended the text consequently.*

12 mostly shallow landslides
*Done.*

13 write 4 and 2 instead of four and two
*Done.*

22 delete "of landslides"
*Done.*

**Page 6**

17 of cumulated
*Done.*

18 of rainfall ..... exceeded...
*Done.*

**Page 7**

1-3 you use 3 times geo-hydrological! use hydrogeological (possibly) and delete sometimes (just use hazards)
*Done.*

6 Probability (E-NEP) algorithm ... record (delete s)
*Done.*

10 and d the time...
*Done.*

18 delete "landslides..... hazards"
*We decided to keep it.*

28 delete else (first word) and write otherwise
*Done.*

**Page 8**

2 possible hazard occurrence ... was calculated (delete using the approach.... (2012)
*Done.*

**Page 9**

6 The analysis ...... DNEPmax is of interest
*Done.*

22 rise following
*Done.*

27 (E-NEP)
*Done.*

32 I, on their own, were not....

*Done.*

**Page 10**

15 et al., 2007
*Done.*

30 delete 2003 (not needed) and De Graff et al., 2013 These are difficult-to-find papers.
*Done.*

32 delete "that lays.... E-NEP."
*Done.*

**Page 11**

2 delete (Brunetti...... 2012)
*Done.*

7 driven hazards
*Done.*

10 these hazards
*Done.*

30 Apulia
*As above.*

**Page 12**

10 (E-NEP)
*Done.*

11 sinkholes). insert space For...
*Done.*

20 events they are abundant
*Done.*

21 as for the
*Done.*

**Page 14**

9 Szonyi
*Done.*

**Page 17**

5 sea level
*Done.*

**Page 18**

I would have placed the rainfall station in a more logical sequence (from N to S? Or from E to W). Now they are placed rather randomly.

*Done.*

2 1-6

*Done.*

2 scheme

*Done.*

The symbols of landslide are placed in ace rain time interval rather precisely, although you stated that only 9 were known to have occurred at precise intervals. Is this just a graphical representation? Or do you ONLY mention the 9 known ones. Explain please.

*We have temporal information only about 9 landslides (4 in cluster A, 1 in cluster B, 1 in cluster C, 3 in cluster D) but we assume these landslides to be representative of the clusters (this is explained in Section 3.3). The landslide occurrence time is provided with an estimated uncertainty represented by the shadowed gray band in Figure 10. We modified the caption to explain it better. In particular, we wrote: "
[revised manuscript text omitted]

[Figure]

| MEAN RAINFALL INTENSITY (mm h⁻¹) | CUMULATED RAINFALL (mm) | TOTAL CUMULATED RAINFALL (mm) | INVENTORY MAP |
|---|---|---|---|

mm h⁻¹
0  3  6  9  12

mm
0  25  50  100  200  500

mm
0  25  50  100  200  500

▲ rain gauge
● landslide in the period
● landslide in previous periods

[Figure]

**Figure 7: Analysis of the spatial and temporal distribution of the event rainfall, and of the triggered event landslides. See text for explanation.**

[Figure]

Figure 8: Logical scheme for the E-NEP algorithm. *D*, rainfall duration, in hours. *E*, cumulated rainfall, in mm. *T*, maximum length of the considered rainfall period, in hours. *d*, time step used to increment the duration of the rainfall period, up to *T*, in hours. *z*, time interval before the next set of (*D,E*) pairs is computed, in hours. {NEP}, set of non-exceedance probability (NEP) values obtained for each (*D,E*) pair adopting the method proposed by Brunetti et al. (2010) and Peruccacci et al. (2012). Statistics of {NEP} are 10th, 25th, 50th, 75th, and 90th percentiles. and $NEP_{max}$ is the maximum value of NEP. $D_{NEPmax}$, event rainfall duration corresponding to $NEP_{max}$. External (blue) loop is run every *z* hours, or fraction of hour. Internal (orange) loop runs from *d* to *T*, in *d* time steps. In Figs. 9, 10 *z* and *d* were set to 1 hour and *T* to 96 hours. See text for explanation.

[Figure]

**Figure 9: Exemplification of the E-NEP algorithm used to provide a non-exceedance probability (NEP) of possible landslide occurrence. (A), (B), (C) show the same rainfall record, and three antecedent conditions corresponding to durations of (A) $D = 6$ hours (red bars), (B) $D = 16$ hours (green bars), and (C) $D = 44$ hours (yellow bars). (D) Shows rainfall $(D,E)$ pairs (grey dots); red, green yellow dots represent the $(D,E)$ pairs corresponding to the conditions shown in (A), (B), and (C). Blue squares show the corresponding non-exceedance probabilities (NEP). Green square represents $NEP_{max}$. See text for explanation.**

[Figure]

Figure 10: Results of the application of the E-NEP algorithm to synthetic rainfall records reconstructed for the locations of four landslide clusters. (A) landslide cluster A; (B) landslide cluster B; (C) landslide cluster C; (D) landslide cluster D. Each panel shows, from top to bottom: (1) hourly rainfall record (blue bars) and cumulated event rainfall (red line); (2) median, E-NEP$_{50}$ (orange line) and maximum, E-NEP$_{max}$ (purple line) values of the non-exceedance probability; ranges E-NEP$_{25}$ – E-NEP$_{75}$ (dark blue shade) and E-NEP$_{10}$ – E-NEP$_{90}$ (light blue shade); (3) rainfall duration corresponding to the most-critical rainfall condition, D-NEP$_{max}$ (green line). The period of occurrence of the landslides (identified by the landslide road sign) is shown by a grey shaded area. The occurrence time (and the associated uncertainty) of nine landslides (cfr. section 3.3) is used to define the landslide

**occurrence period of the four clusters.** The time of occurrence of peak flow (identified by the flood road sign) is shown by the vertical blue line. See text for explanation.